# An atomic-resolution view of neofunctionalization in the evolution of apicomplexan lactate dehydrogenases

Jeffrey I Boucher[1], Joseph R Jacobowitz[1], Brian C Beckett[1], Scott Classen[2], Douglas L Theobald[1]*

[1]Department of Biochemistry, Brandeis University, Waltham, United States; [2]Physical Biosciences Division, Lawrence Berkeley National Laboratory, Berkeley, United States

**Abstract** Malate and lactate dehydrogenases (MDH and LDH) are homologous, core metabolic enzymes that share a fold and catalytic mechanism yet possess strict specificity for their substrates. In the Apicomplexa, convergent evolution of an unusual LDH from MDH produced a difference in specificity exceeding 12 orders of magnitude. The mechanisms responsible for this extraordinary functional shift are currently unknown. Using ancestral protein resurrection, we find that specificity evolved in apicomplexan LDHs by classic neofunctionalization characterized by long-range epistasis, a promiscuous intermediate, and few gain-of-function mutations of large effect. In canonical MDHs and LDHs, a single residue in the active-site loop governs substrate specificity: Arg102 in MDHs and Gln102 in LDHs. During the evolution of the apicomplexan LDH, however, specificity switched via an insertion that shifted the position and identity of this 'specificity residue' to Trp107f. Residues far from the active site also determine specificity, as shown by the crystal structures of three ancestral proteins bracketing the key duplication event. This work provides an unprecedented atomic-resolution view of evolutionary trajectories creating a nascent enzymatic function.

*For correspondence: dtheobald@brandeis.edu

Competing interests: The authors declare that no competing interests exist.

## Introduction

The common ancestor of the eukaryotic Apicomplexa evolved nearly 1 billion years ago (**Douzery et al., 2004**), and its modern descendants comprise a large phylum of intracellular parasites that are currently responsible for numerous devastating metazoan diseases, including malaria (*Plasmodium*), toxoplasmosis (*Toxoplasma*), cryptosporidiosis (*Cryptosporidium*), cyclosporiasis (*Cyclospora*), and babesiosis (*Babesia*). A key event in the early evolution of the Apicomplexa was the acquisition of a malate dehydrogenase (MDH) via lateral gene transfer from α-proteobacteria (**Golding and Dean, 1998**; **Madern, 2002**; **Zhu and Keithly, 2002**). Following a gene duplication event roughly 700–900 Mya, one copy of this MDH evolved a novel substrate specificity to become a highly specific lactate dehydrogenase (LDH) that is now essential to the life cycle of many modern apicomplexans (**Royer et al., 1986**). As a core metabolic enzyme that evolved independently of metazoan LDH, the unique apicomplexan LDH has attracted significant attention as a potential drug target (**Gomez et al., 1997**; **Read et al., 1999**; **Cameron et al., 2004**; **Conners et al., 2005**). However, the molecular and evolutionary mechanisms that drove this switch in substrate specificity are currently unknown.

LDH and MDH are homologous, 2-ketoacid oxidoreductases that share both a protein fold (**Rossmann et al., 1975**; *Figure 1—figure supplement 1*) and a common catalytic mechanism (**Birktoft and Banaszak, 1983**; **Clarke et al., 1986**; **Hart et al., 1987a, 1987b**; **Clarke et al., 1988**; **Waldman et al., 1988**; *Figure 1*). Both enzymes are found in central metabolism: MDH catalyzes the interconversion of oxaloacetate and malate in the citric acid cycle, and LDH converts pyruvate to lactate in the

**eLife digest** How are new genes created? Most of the mutations in the genome of an organism place the organism at some sort of disadvantage, but a small number confer an advantage. The beneficial changes are usually retained by subsequent generations and can ultimately lead to the creation of new genes.

An example is the gene that encodes an enzyme called lactate dehydrogenase (LDH). This enzyme is involved in anaerobic respiration, the process that allows organisms to produce energy without using oxygen. The LDH enzyme is found in many species of animals and parasites, including those that spread malaria and other diseases. However, there are important differences in the structures of the LDH enzyme in animals and some parasites, like the malarial *Plasmodium*, because the genes for the enzymes in these two groups evolved separately.

The parasite version of the LDH enzyme evolved hundreds of millions of years ago from an enzyme with a similar structure called malate dehydrogenase, which was inherited from bacteria. To work out how the LDH enzyme developed, Boucher et al. predicted and built the ancestral proteins that would have formed as the bacterial enzyme evolved into LDH.

Studying these structures revealed that two mutations were mainly responsible for this evolution: six amino acids were added to the active site of the enzyme, and one amino acid (at position 102) was replaced by a different amino acid. However, introducing the same mutations into a modern version of the bacterial enzyme did not produce a working form of the LDH enzyme. This suggests that other amino acids, further away from the active site, also influenced how LDH evolved.

The structures found by Boucher et al. reveal that the enzymes evolved as a result of a gene duplicating, followed by one of the copies evolving a new function. However, some of the mutations responsible for the novel function occurred far from the active site, and it is still unknown how they exert their functional effects. Untangling the important mutations from the mundane will be necessary to fully understand how protein functions are created and how to control them—both of which will aid in developing effective drugs that target essential parasite proteins.

final step of anaerobic glycolysis. Despite their structural and catalytic similarities, modern apicomplexan LDHs and MDHs have extraordinarily strict substrate specificity. For example, *Plasmodium falciparum* (*Pf*) MDH and LDH each prefer their respective substrates by over six orders of magnitude. The biophysical basis for this extraordinary substrate preference is presently an unresolved question.

A conspicuous structural difference between apicomplexan MDHs and LDHs is an insertion within the active site loop of the LDHs (*Bzik et al., 1993*; *Dunn et al., 1996*; *Figure 2*). In the LDH/MDH superfamily, closure of this loop over the active site is rate-limiting during catalysis (*Waldman et al., 1988*), and mutations within this loop have large effects on activity and substrate specificity (*Wilks et al., 1988*). For example, simply mutating Gln102 to Arg in the specificity loop of *Bacillus stearothermophilus* (*Bs*) LDH converts the enzyme into an MDH, shifting specificity from a $10^3$-fold preference for pyruvate to a $10^4$-fold preference for oxaloacetate (*Wilks et al., 1988*) (*Figure 1*, residue numbering is based on the dogfish LDH convention [*Eventoff et al., 1977*]). In fact, all known MDHs have an Arg at position 102, while canonical LDHs have a Gln, and consequently residue 102 has been called the 'specificity residue' (*Chapman et al., 1999*). Residue 102 is thought to contribute to substrate discrimination by balancing the substrate charge within the active site: the positively charged Arg in MDHs forms a salt bridge with the C4 carboxylate of oxaloacetate, whereas the neutral Gln in canonical LDHs packs with the C3 methyl of pyruvate (*Figure 1*). Yet, attempts to convert an MDH into an LDH by mutating Arg102 to Gln have met with limited success (*Nicholls et al., 1992*; *Cendrin et al., 1993*). In the apicomplexan LDHs, residue 102 is not a Gln but a Lys, a relatively conservative substitution compared to the MDH Arg. It is currently not understood why *Plasmodium* LDHs lack activity towards oxaloacetate, despite having a positively charged sidechain at residue 102 similar to MDHs (*Gomez et al., 1997*; *Dando et al., 2001*; *Winter et al., 2003*; *Brown et al., 2004*; *Kavanagh et al., 2004*; *Shoemark et al., 2007*).

Apicomplexan LDH evolved from the duplication of an ancestral MDH gene (*Golding and Dean, 1998*; *Zhu and Keithly, 2002*). Gene duplication is widely considered the major force that has driven the evolutionary diversity of protein functions (*Innan and Kondrashov, 2010*). There are three general

**Figure 1**. Schematic of M/LDH superfamily active site and catalytic mechanism. MDH reduces oxaloacetate to malate, in which the R-group is a methylene carboxylate group. LDH reduces pyruvate to lactate, in which the R-group is a methyl group. Key conserved active site residues are shown in black; substrate is shown in blue. The oxidized 2-ketoacid form of the substrate is at left; the reduced 2-hydroxy acid form is shown at right. The R-group of the substrate interacts with Arg102 in MDHs and Gln102 in canonical LDHs. Both Arg109 and position 102 are found in the mobile 'specificity loop' that closes over the active site.

The following figure supplements are available for figure 1:

**Figure supplement 1**. Fold architecture in the LDH and MDH superfamily.

ways duplicated genes can be fixed in a population by selection: (1) 'dosage selection', beneficial increase in dosage due to multiple copies, (2) 'subfunctionalization', specialization of previously existing functions, or (3) 'neofunctionalization', creation of a novel function through the accumulation of beneficial, gain-of-function mutations (*Ohno, 1970*). Most mutations, however, are either neutral or detrimental. A new duplicated gene typically degrades to a crippled pseudogene before it can acquire the rare beneficial mutations needed to confer a selectable function (*Walsh, 1995*; *Lynch and Conery, 2000*). Hence, classical neofunctionalization has fallen out of favor in preference for models that begin with the duplication of a multifunctional protein, such as 'specialization' and 'subfunctionalization' models. Currently the molecular and evolutionary mechanisms that create novel functions in gene duplicates are fiercely debated (*Force et al., 1999*; *Conant and Wolfe, 2008*; *Innan and Kondrashov, 2010*; *Soskine and Tawfik, 2010*), and there are few clear examples of classic neofunctionalization or gain-of-function mutations (*Zhang and Rosenberg, 2002*; *Bridgham et al., 2008*; *Voordeckers et al., 2012*).

The apicomplexan LDH and MDH enzyme family provides an exceptional model system for investigating several long-standing questions in molecular evolution, including the mechanisms available to convergent evolution, the number of mutations required to produce a nascent function, the role of promiscuous intermediates during evolution of function, and the effects of epistasis on evolutionary irreversibility. In order to identify the biophysical and evolutionary mechanisms responsible for pyruvate specificity in apicomplexan LDHs, we have reconstructed ancestral proteins along the evolutionary trajectories leading to modern apicomplexan MDHs and LDHs (*Figure 3B*). We kinetically and structurally characterized the ancestral proteins together with multiple evolutionary intermediates. This work provides a clear example of neofunctionalization in protein evolution and the first crystal structures documenting the evolution of a new enzyme. We show that apicomplexan LDHs evolved as the result of few mutations of large effect via the classic neofunctionalization of a duplicated MDH gene.

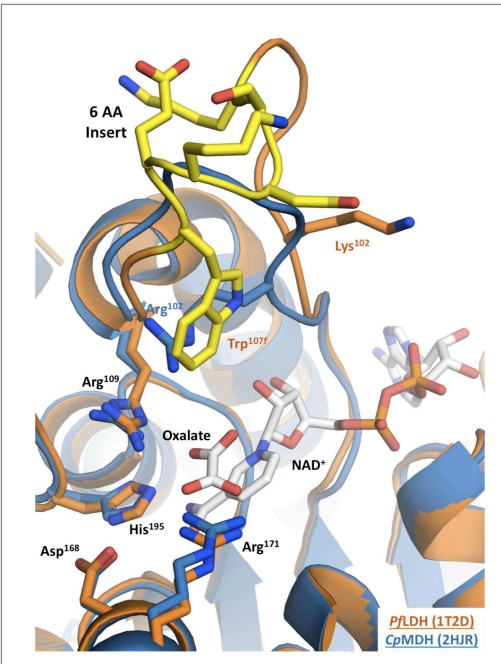

**Figure 2**. Apicomplexan M/LDH active sites. Structures of *Cp*MDH (blue, 2hjr) and *Pf*LDH (vermilion, 1t2d) superposed using THESEUS. The ligands (oxalate and NAD+) are from 1t2d and colored white. Side chains of important residues are shown as sticks and the six-residue insert of *Pf*LDH is highlighted in yellow. Note how the *Pf*LDH Trp107f overlays Arg102 from *Cp*MDH. Residues in the insertion are labeled using numbers and letters to maintain consistency with homologous positions in the dogfish LDH.

The following source data and figure supplements are available for figure 2:

**Figure supplement 1**. Sequence alignment of the specificity loop from apicomplexan M/LDHs with ancestral sequences.

**Figure supplement 2**. Alanine scanning of *Pf*LDH specificity loop.

**Figure supplement 2—source data 1**. Kinetic parameters for PfLDH alanine-scan.

**Figure supplement 3**. Crystal structure of *Pf*LDH-W107fA mutant.

# Results

## LDH enzymes have evolved independently at least four times

A maximum likelihood phylogeny of representatives of all known LDH and MDH proteins provides strong support for five distinct protein clades (**Figure 3A**, **Figure 3—figure supplement 1**): canonical LDHs, 'LDH-like' MDHs, mitochondrial-like MDHs, cytosolic-like MDHs, and the poorly characterized HicDHs (hydroxyisocaproate-related dehydrogenases), confirming previous phylogenetic analyses (**Golding and Dean, 1998**; **Madern, 2002**; **Zhu and Keithly, 2002**; **Madern et al., 2004**).

The HicDH clade are close sequence homologs of a known hydroxyisocaproate dehydrogenase. They all possess a residue other than a Gln or an Arg at the 'specificity' position 102, as well as insertions of varying lengths within the catalytic loop between residues 102 and 109. Despite these alterations within the catalytic loop, all other catalytic residues (Arg109, Asp168, Arg171, and His195) are conserved. Only one taxon within the HicDH clade has been functionally characterized, DHL2_LACCO, which is a specific hydroxyiso-caproate dehydrogenase (**Feil et al., 1994**). These observations suggest that the clade features dehydrogenases with altered substrate specificity.

Except for the HicDHs, which are exclusively eubacterial, both eukaryotic and eubacterial enzymes are found in all major clades (**Figure 3—figure supplement 2**). The 'LDH-like' MDH clade additionally contains archaeal dehydrogenases, which are basal and group to the exclusion of the bacterial MDHs.

Intriguingly, three different groups of LDH proteins cluster with high confidence outside of the canonical LDH clade. A set of trichomonad LDHs found in the cytosolic-like MDH clade are thought to have evolved from a recent gene duplication of an MDH (**Wu and Fiser, 1999**). The Trichomonads appear to lack a canonical LDH. A prominent eukaryotic group of LDH and MDH proteins from the Apicomplexa nests deep within the bacterial 'LDH-like' MDHs, sister to many Rickettsiales sequences, signifying a horizontal gene transfer event from α-proteobacteria to the eukaryotic Apicomplexa. We find no evidence that the Apicomplexa have canonical LDH or conventional eukaryotic-type MDH (either cytosolic- or mitochondrial-like MDHs), despite searching in many available complete apicomplexan genomes (multiple Eimeria, Neospora, Toxoplasma, Plasmodium, and Cryptosporidium species) (**Heiges et al., 2006**; **Gajria et al., 2008**; **Aurrecoechea et al., 2009**). In the Apicomplexa, LDH activity has apparently evolved independently twice (**Figure 3B**, **Figure 3—figure supplement 3**), once in a lineage leading to *Plasmodium*-related species and once in *Cryptosporidium*. The apicomplexan portion of the LDH/MDH gene phylogeny is consistent with recent apicomplexan species phylogenies constructed from concatenated protein sequences (**Templeton et al., 2009**).

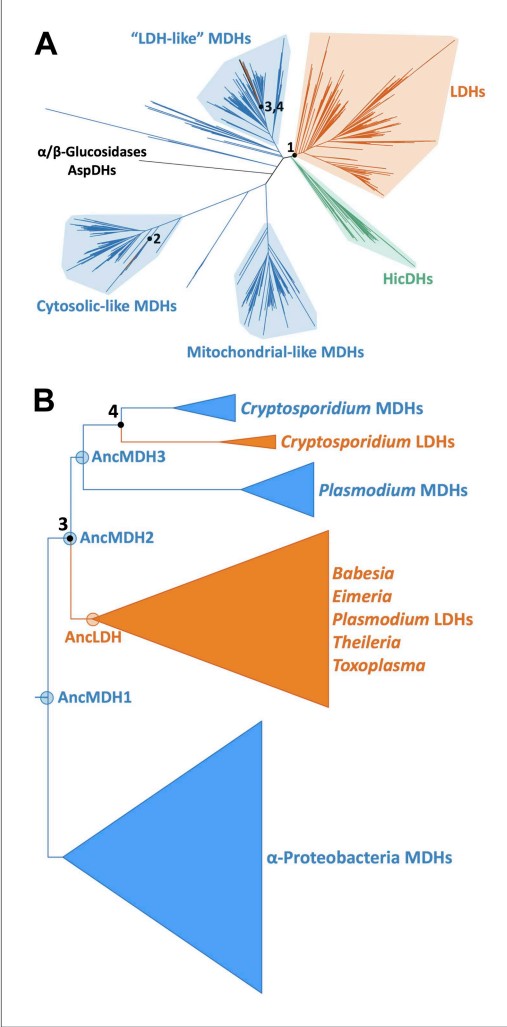

**Figure 3**. Phylogeny of apicomplexan M/LDH superfamily. (**A**) 1844 taxa. The tree is colored according to function (LDH—vermilion; MDH—blue; HicDH—moss). The N-terminal Rossmann-fold of glucosidases and aspartate dehydrogenases (AspDHs) was used to root the phylogeny. Numbers highlight convergent events of LDH evolution from MDHs: 1–Canonical LDHs, 2–Trichomonad LDHs, and 3,4–apicomplexan LDHs. The shaded clades have highly significant supports (**Anisimova and Gascuel, 2006**). (**B**) Apicomplexan M/LDH Clade. A close-up of the apicomplexan portion of the phylogeny in **A**, similarly colored by function. aLRT supports for each group: α-proteobacteria MDHs, 15; apicomplexan LDHs, 11; *Plasmodium* LDHs, 333; *Cryptosporidium* MDHs, 54; *Cryptosporidium* LDHs, 202. Ancestral reconstructed proteins are labeled at internal nodes (AncMDH1, AncMDH2, AncMDH3, AncLDH). The focus of the present work is the gene duplication at node 3.

The following figure supplements are available for figure 3:

**Figure supplement 1**. Phylogeny of M/LDH superfamily.

*Figure 3. Continued on next page*

We rooted the MDH/LDH phylogeny using the Rossmann fold domain of the distantly related α/β-glucosidases and aspartate dehydrogenases as outgroups. The ML root position apparently splits the tree into two large groups: one which contains the cytosolic- and mitochondrial-like MDHs, which are largely dimeric, and another which contains the canonical LDHs, 'LDH-like' MDHs, and HicDHs, which are primarily tetrameric (*Figure 3—figure supplement 1*). While the ML root position is robust to variation in taxon coverage, the exact location is poorly supported. Nevertheless, there is strong support for a root position within the central MDH section of the tree and outside of the five identified clades, including the canonical LDH clade (confidence level >0.99985 according to the aLRT), indicating that the canonical LDHs evolved from an ancestral MDH. The global rooting and the location of the three separate LDH groups, deep within MDH clades, indicate that LDH enzymes have evolved convergently from MDHs at least four times in the superfamily.

## An insertion in the catalytic loop of apicomplexan LDHs

In the present work, our focus is on the convergent evolution of the unusual apicomplexan LDHs. With the α-proteobacterial 'LDH-like' MDHs as the closest outgroup, the apicomplexan enzymes are split into two main groups: LDHs belonging to *Toxoplasma*, *Plasmodium*, and related protists, and MDHs belonging to *Plasmodium* and *Cryptosporidium*. Apart from their atypical phylogenetic position, the apicomplexan MDHs appear as typical α-proteobacterial 'LDH-like' MDHs, containing all the key catalytic residues including Arg102. The *Cryptosporidium* LDHs are an exception, being nested within the apicomplexan MDH clade partitioned from the rest of the apicomplexan LDHs. *Cryptosporidium* LDHs have a Gly at position 102 and are thought to be a product of an independent, convergent duplication event (*Madern et al., 2004*).

In contrast, the large apicomplexan LDH clade is demarcated by a unique, conserved five-residue insertion in the active site loop. While the apicomplexan LDH and MDH proteins are moderately divergent, with about 45% sequence identity, the differences are largely confined to exterior residues removed from the active sites. One important difference is that the apicomplexan LDHs have Lys102 for the 'specificity residue', rather than a Gln as found in the canonical LDHs (*Figure 2— figure supplement 1*). Apicomplexan proteins frequently contain numerous insertions relative to

Figure 3. Continued

**Figure supplement 2**. Phylogeny of M/LDH superfamily.

**Figure supplement 3**. Apicomplexan M/LDH Clade.

proteins from other species (*Feng et al., 2006*; *Kissinger and DeBarry, 2011*), a characteristic thought to result from various factors, including high AT genome content, DNA strand slippage, double strand break repair, high recombination rates, and selection pressure for parasite antigenic variation. Except for Met106, the amino acid and coding sequence immediately flanking the apicomplexan LDH loop insertion is largely conserved with α-proteobacterial MDHs (*Figure 2—figure supplement 1*). It is therefore likely that a mutation 'expanded' the Met106 codon to code for six residues, resulting in the observed five-residue insertion and the Met106Lys mutation. Henceforth we will refer to this expansion mutation as the 'six-residue loop insertion'.

## Trp107f is the modern apicomplexan LDH specificity residue

In the modern apicomplexan enzymes, the six-residue insertion in the LDH specificity loop (positions 99–112) induces two significant structural changes relative to MDH (*Figure 2*). First, LDH residue Lys102 is excluded from of the active site, unlike the corresponding Arg102 in MDH, which is enclosed within the active site and participates in functionally important interactions with the substrate. Second, LDH Trp107f, which is part of the novel insertion, occupies the same space as Arg102 in MDH (by convention, residues in the insertion are labeled using numbers and letters to maintain consistency with homologous positions in the dogfish LDH, *Figure 2*).

The only prominent structural difference between the active sites of the LDH and MDH proteins is the replacement of MDH Arg102 with LDH Trp107f. Trp107f is positioned where it could presumably interact with the distinguishing C3 methyl of the pyruvate substrate, while MDH Arg102 interacts with the C4 carboxylate of oxaloacetate (*Chapman et al., 1999*). As a bulky, hydrophobic residue, Trp107f could recognize pyruvate in preference to oxaloacetate by two mechanisms: a hydrophobic interaction with the pyruvate C3 methyl vs the negatively charged oxaloacetate methylene carboxylate and steric occlusion of the methylene carboxylate of oxaloacetate. Furthermore, Trp107f is conserved in all apicomplexan LDHs (*Figure 2—figure supplement 1*), suggesting negative selection and functional importance. We therefore hypothesized that Trp107f plays an important role in pyruvate recognition.

We tested the functional importance of residues in the specificity loop in *Pf*LDH with an 'alanine scan' by individually mutating each residue in positions 101–108 to an alanine (*Figure 2—figure supplement 2*, note Ala103 was mutated to a serine). We assessed the activity of the mutants using $k_{cat}/K_m$, a measure of enzymatic specificity and catalytic efficiency, as determined from steady state kinetic assays. Mutating Trp107f to Ala reduced pyruvate activity by five orders of magnitude, whereas mutations at all other positions had effects less than a single order of magnitude, including the canonical specificity residue at position 102. The Trp107fAla mutation affects both $k_{cat}$ (1500-fold decrease) and $K_m$ (50-fold increase).

To assess the effects of Trp107fAla mutation on the specificity loop conformation, we solved the crystal structure of *Pf*LDH-W107fA (1.1 Å ) in the presence of oxamate and NADH. The protein crystallizes in the same space group as the wild-type *Pf*LDH, with nearly identical cell dimensions (*Figure 2—figure supplement 3A*). In the W107fA mutant, the specificity loop is disordered between residues Thr101 and Arg109, as is often seen in structures in which the loop is in the open conformation. In the mutant, residues 112–115 are in a linear α-helical conformation, in contrast to the wild-type *Pf*LDH closed state which has a very prominent 60° kink in the α-helix at Pro114. Thus, the only significant difference between the wild-type and mutant structures is that the *Pf*LDH-W107fA specificity loop is found in the open conformation, consistent with weaker binding of substrate (*Figure 2—figure supplement 3B*). These results indicate that Trp107f is necessary for pyruvate activity in apicomplexan LDHs, and that it has become the new 'specificity residue' despite the fact that Trp107f does not align in sequence with the canonical specificity residue at position 102 (*Figure 2—figure supplement 1*).

## The loop insert fails to swap specificity in modern LDH and MDH

During evolution, the six-residue insertion displaced the canonical specificity residue at position 102 and apparently switched substrate preference in apicomplexan LDHs. If this insertion is sufficient for pyruvate recognition, then adding the insertion to a modern apicomplexan MDH should convert the enzyme to an LDH. To test this hypothesis, we incorporated the six-residue insertion from *Pf*LDH into the catalytic loop

of *Pf*MDH (*Pf*MDH-INS) and the *Cryptosporidium parvum* (*Cp*) MDH (*Cp*MDH-INS). The chimeric proteins showed a >100-fold reduction in oxaloacetate activity with no significant gain in pyruvate activity (*Figure 4*). Like other MDHs, the apicomplexan MDHs have an Arg at position 102 that is important for oxaloacetate recognition; in the modern apicomplexan LDHs position 102 is a Lys. The Arg102Lys mutation may be necessary to eliminate oxaloacetate activity and increase pyruvate activity. Therefore, we also mutated Arg102 to Lys in the *Pf*MDH chimera (*Pf*MDH-R102K-INS). However, this mutation reduced activity towards oxaloacetate by another 100-fold, with no increase in pyruvate activity (*Figure 4*).

Alternatively, it may be possible to revert a modern apicomplexan LDH to MDH-like specificity by deleting its six-residue loop insertion. To test this hypothesis we removed the insertion from *Pf*LDH and from the *Toxoplasma gondii* (*Tg*) LDH2 (constructs *Pf*LDH-DEL and *Tg*LDH2-DEL). However, deleting the insertion from the modern LDHs abolishes pyruvate activity with no significant gain of oxaloacetate activity (*Figure 4*). Both of these deletion mutants retain a Lys at position 102, but a specific MDH likely requires an Arg at position 102. Mutating Lys102 to Arg in *Pf*LDH-DEL results in a two order-of-magnitude gain in oxaloacetate activity (*Figure 4*). However, this mutant fails to recapitulate the level of oxaloacetate activity seen in modern apicomplexan MDHs. In the modern enzymes, substrate specificity cannot be switched with mutations involving the loop insert and position 102, indicating that additional residues govern substrate preference.

## The ancestral MDH and LDH enzymes are specific and highly active

The apicomplexan LDH and MDH phylogeny strongly suggests that after (or coincident with) the crucial gene duplication event, the nascent LDH branch gained pyruvate activity due to the six-residue insertion in the specificity loop. This presents a conundrum, as our mutation trials in the modern enzymes failed to recapitulate the historical swap in specificity. However, the modern apicomplexan LDH and MDH enzymes differ by over 200 residues in addition to the loop insert and Arg102Lys, differences that have accumulated in the descendants of the ancestral MDH and LDH. Any of these differences may detrimentally affect the ability to switch substrate specificity with the insertion in the modern enzymes. We therefore reasoned that the ancestral background may be necessary for swapping specificity with the loop insertion. To test this, we reconstructed and characterized four key ancestral enzymes: AncMDH1, the ancestral protein that was transferred from α-proteobacteria to the archaic Apicomplexa, AncMDH2, the last common ancestor of all apicomplexan MDHs and LDHs, found at the critical duplication event, AncMDH3, the last common ancestor of all modern apicomplexan MDHs, and AncLDH, the last common ancestor of modern apicomplexan LDHs (*Figure 3B*).

All four ancestral proteins are highly active in steady state kinetic assays, with substrate preferences and catalytic efficiencies that are similar to their modern apicomplexan descendants (*Figure 5*), despite sharing only 49–71% sequence identity with the modern apicomplexan proteins (*Figure 5—figure supplement 1*). AncMDH1, AncMDH2, and AncMDH3 are highly specific MDHs with negligible pyruvate activity, having even greater activity towards oxaloacetate than modern *Plasmodium* and *Cryptosporidium* MDHs (*Figure 5*). AncLDH is a highly active and specific LDH, with very low activity towards oxaloacetate (*Figure 5*).

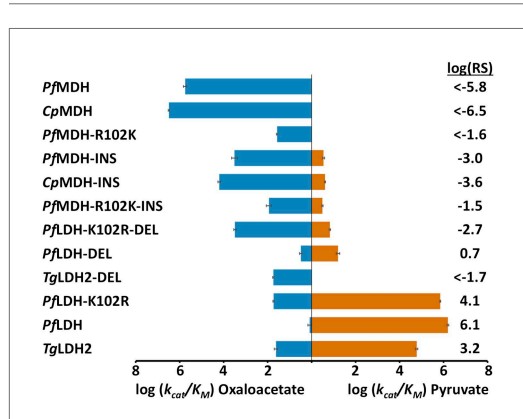

**Figure 4**. Specificity switching in apicomplexan M/LDHs. Blue horizontal bars (left) quantify activity towards oxaloacetate; vermilion horizontal bars (right) quantify activity towards pyruvate. Activity is measured as $\log_{10}(k_{cat}/K_M)$, where $k_{cat}/K_M$ is in units of $s^{-1}M^{-1}$. Error bars are shown as small black brackets and represent 1 SD of the mean from triplicate measurements. INS refers to the presence of the six-residue insertion from *Pf*LDH, DEL refers to the removal of the six-residue insertion. Relative specificity (RS) is the ratio of $k_{cat}/K_M$ for pyruvate vs oxaloacetate, with positive $\log_{10}(RS)$ representing a preference for pyruvate and negative $\log_{10}(RS)$ representing a preference for oxaloacetate. All logarithms are base 10.

The following source data are available for figure 4:

**Source data 1**. Kinetic parameters for modern constructs.

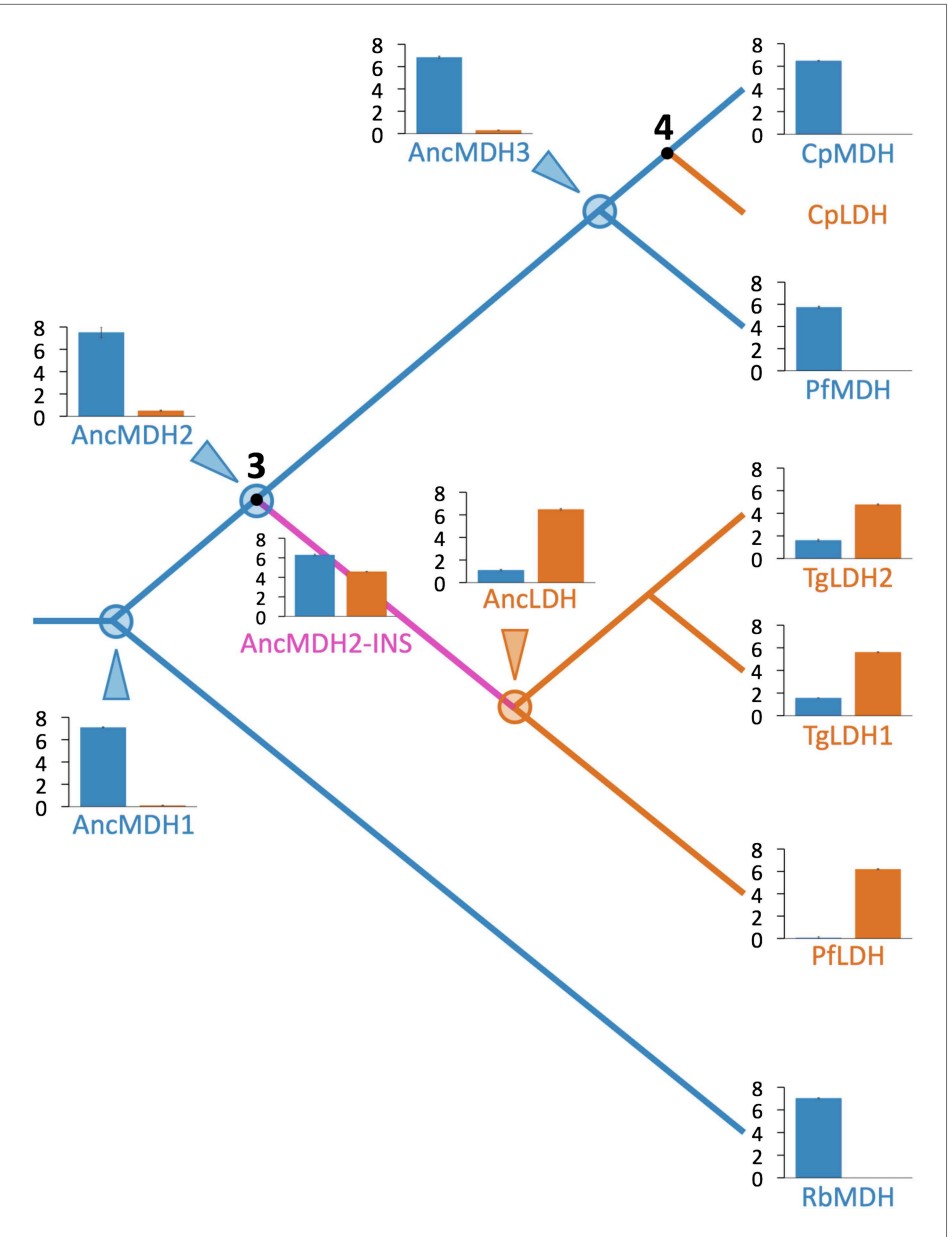

**Figure 5**. Evolution of novel LDHs in Apicomplexa. The activities of ancestral and modern apicomplexan M/LDHs are plotted on the corresponding nodes of the protein phylogeny. Nodes are numbered as in **Figure 3B**. The y-axis of the bar graphs is log($k_{cat}/K_M$), with oxaloacetate in blue and pyruvate in vermilion. *Rb*MDH is a representative α-proteobacterial MDH from *Rickettsia bellii*. *T. gondii* has two LDH proteins (TgLDH1 and TgLDH2), each expressed at different stages of the life cycle (***Dando et al., 2001***).

The following source data and figure supplements are available for figure 5:

**Source data 1**. Kinetic parameters for ancestral/modern phylogeny.

**Figure supplement 1**. Sequence identity of ancestral and modern proteins.

## The loop insert successfully swaps specificity in both ancestral LDH and MDH

AncLDH differs from AncMDH2 by 66 residues, including the six-residue insertion and Arg102Lys. We investigated the evolutionary trajectory from AncMDH2 to AncLDH by characterizing three different

mutations in the AncMDH2 background: the addition of AncLDH's six-residue insertion to the AncMDH2 specificity loop, Arg102Lys, which assesses the effect of changing the canonical specificity residue, and the remaining 59 residues that separate AncLDH from AncMDH2, simultaneously changed to their AncLDH identities.

Incorporating the loop insertion into AncMDH2 confers significant pyruvate activity with minimal effect on oxaloacetate activity, resulting in a highly active, bifunctional enzyme (AncMDH2-INS, *Figure 6*). In contrast, the Arg102Lys mutation in the AncMDH2 background (AncMDH2-R102K, *Figure 6*) reduces oxaloacetate activity by more than a 100-fold, with no increase in pyruvate activity. The 59 mutations in the AncMDH2 background have a minimal effect on the activity towards both substrates (AncMDH2-59Mut, *Figure 6*). Note that the AncMDH2-59Mut construct is equivalent to a modified AncLDH construct with the Lys102Arg mutation and the insertion deleted from the loop. Therefore, only two changes—Lys102Arg and the loop deletion—are sufficient to convert the AncLDH construct to a highly active and specific MDH.

Combinations of these mutations confirm that the insertion is primarily responsible for the evolution of pyruvate activity. Adding the 59 mutations to AncMDH2-INS (resulting in a construct that differs from AncLDH by only one residue) has little additional effect (AncMDH2-INS-59Mut, *Figure 6*). Surprisingly, the combination of Arg102Lys and the 59 mutations, a construct that differs from AncLDH by just the six-residue insertion, yields a crippled MDH enzyme with 1000-fold less oxaloacetate activity than AncMDH2 (AncMDH2-R102K-59Mut, *Figure 6*). However, the combination of Arg102Lys and the loop insertion in the AncMDH2 background is sufficient to confer pyruvate activity and specificity comparable to AncLDH (AncMDH2-INS-R102K, *Figure 6*).

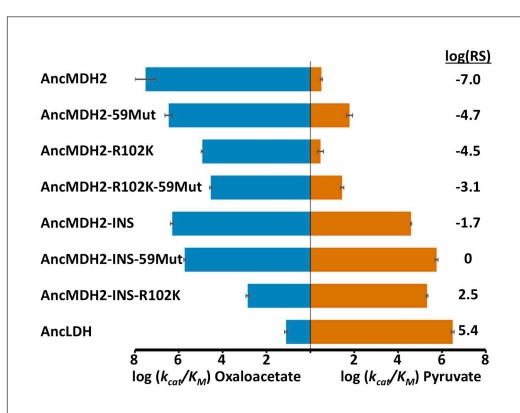

**Figure 6**. Specificity switching in ancestral MDH2. INS refers to the reconstructed six-residue insertion from AncLDH. 59Mut is described in the text. Relative specificity (RS) is described in legend of *Figure 4*.

The following source data and figure supplements are available for figure 6:

**Source data 1**. Kinetic parameters for ancestral specificity switch mutants.

**Source data 2**. Source data for figure supplement 7. Kinetic parameters for alternative ancestral proteins.

**Figure supplement 1**. Histogram of ancestral reconstructions.

**Figure supplement 2**. Histogram of ancestral reconstructions.

**Figure supplement 3**. Histogram of ancestral reconstructions.

**Figure supplement 4**. Histogram of ancestral reconstructions.

**Figure supplement 5**. Histogram of ancestral reconstructions.

**Figure supplement 6**. Histogram of ancestral reconstructions.

**Figure supplement 7**. Alternative ancestral enzymes.

## Ancestral kinetics are robust to reconstruction uncertainty

Ancestral sequence reconstruction is a difficult statistical problem that strongly relies on evolutionary assumptions, which may be unrealistic, and on available sequence data, which is inherently incomplete. The likelihood and Bayesian ancestral reconstruction methodology that we use produces the most probable ancestral sequence given certain evolutionary model assumptions, along with a posterior probability for alternative amino acids at each position (*Figure 6—figure supplements 1–6*). Ambiguous residues are generally associated with positions of low conservation and presumably less functional importance. The reconstructed AncMDH2 and AncLDH sequences have 31 and 48 ambiguous positions, respectively, all of which are located outside of the 'first active site shell' (defined as within 6 Å of the substrate). In order to verify that these sequence ambiguities do not affect our kinetic results, alternative ancestral sequences were reconstructed and assayed. We tested the robustness of our ancestral proteins by constructing alternative ancestors based on perturbed sequence

data, evolutionary assumptions, and phylogenetic methodology. Both phylogenies give very similar relationships, and *Figure 2B* summarizes both equally well. The alternative AncMDH2 (AncMDH2*) differs from AncMDH2 by 27 residues; the alternative AncLDH (AncLDH*) differs from AncLDH by 19 residues.

The alternative ancestral reconstructions behave very similar to the prior reconstructions. AncMDH2* is a strict MDH, and AncLDH* is a strict LDH (*Figure 6*, *Figure 6—figure supplement 7*). Addition of the six-residue insertion from AncLDH* to AncMDH2* confers pyruvate specificity without adversely affecting oxaloacetate activity (AncMDH2*-INS, *Figure 6—figure supplement 7*). In the AncMDH2* background, mutating Arg102 to Lys together with the 58 mutations from AncLDH* yields a poor enzyme with little pyruvate activity (AncMDH2*-R102K-58Mut). The kinetic behavior of these AncMDH2* constructs closely matches those seen with the corresponding AncMDH2 constructs (AncMDH2, AncMDH2-INS, and AncMDH2-R102K-59Mut, *Figure 6*).

## Crystal structures of ancestral MDH, LDH, and an evolutionary intermediate

In order to understand the structural changes during evolution that shifted the enzymatic substrate specificity of the apicomplexan dehydrogenases, we determined the high-resolution crystal structures of three ancestral proteins bracketing the key duplication event: AncMDH2, AncLDH*, and AncMDH2-INS. Each protein was crystallized with multiple substrates or ligands (lactate, malate, and oxamate inhibitor) and with NADH (resolution ranging from 1.35 Å to 2.05 Å). Unfortunately, in all crystals with malate, the malate spontaneously converted to pyruvate and/or lactate via redox reactions and decarboxylation. In the crystals constructed with AncMDH2 and lactate, a phosphate was seen in the active site rather than lactate. In the following, therefore, the descriptions of the models are primarily based on enzymes crystallized with oxamate inhibitor or lactate, which are highly similar. All three ancestral proteins adopt the same overall fold and conformation as the modern, descendant enzymes. In particular, the ancestral active sites and specificity loops are highly similar to their modern counterparts.

## Ancestral malate dehydrogenase: AncMDH2

The AncMDH2 structure superposes closely with the modern *Cp*MDH structure (*Vedadi et al., 2007*) (~0.6 Å RMSD for the loop-closed states), although differing at ~119 residue positions (62% sequence identity, *Figure 7A*). In the modern and ancestral MDHs, all residues within the first shell of the active sites (within 6 Å of the substrate) are identical, and the active site conformations are correspondingly highly similar (*Figure 7B*). The first shell active site residues comprise Arg102, Arg109, Leu112, Asn140, Leu167, Asp168, Arg171, His195, Met199, Gly236, Gly237, Ile239, Val240, Ser245, Ala246, and Pro250.

Compared to the modern MDH, only slight differences are seen in the substrate loop backbone and the positioning of the Arg102 and Arg109 sidechains, which are the only residues from the specificity loop that directly interact with the substrate. However, these modest conformational differences are largely within coordinate error, as the loop residues have some of the highest B-factors in the structures. Furthermore, AncMDH2 was crystallized with lactate/oxamate and NADH, while *Cp*MDH was crystallized with citrate and ADPR (an NADH analog lacking the nicotinamide ring). Citrate is roughly three times larger than lactate and has likely affected the position of substrate loop in the *Cp*MDH structure.

### Ancestral lactate dehydrogenase: AncLDH*

The ancestral AncLDH* and modern apicomplexan LDH structures are likewise highly similar (*Winter et al., 2003*; *Cameron et al., 2004*; *Kavanagh et al., 2004*) (RMSD ~0.8 Å for the loop-closed states, *Figure 7C*), while sharing only 63–71% sequence identity. The first shell active site residues are identical in the AncLDH* and modern *Toxoplasma* LDHs, comprising the same residues as the apicomplexan MDH active site with the sole exception of position 102, which is replaced by Trp107f in the LDHs. The modern *Plasmodium* LDHs have two different residues in the active site first shell: Pro246 and Ala236, rather than Ala246 and Gly236 as found in *Tg*LDH1, *Tg*LDH2, and AncLDH*. The conformations of the ancestral and modern active sites are nearly indistinguishable, with only small differences in the specificity loop conformation (*Figure 7D*).

In both the ancestral and modern LDH structures, Trp107f and Arg109 are the only residues from the specificity loop that interact with the substrate. As in the modern LDH structures, ancestral Lys102 does not interact with the substrate but points away from the active site into solution. In contrast, Trp107f is buried within the active site, with the edge of the indole ring interacting with the pyruvate C3 methyl, which is the very chemical moiety that distinguishes pyruvate from oxaloacetate.

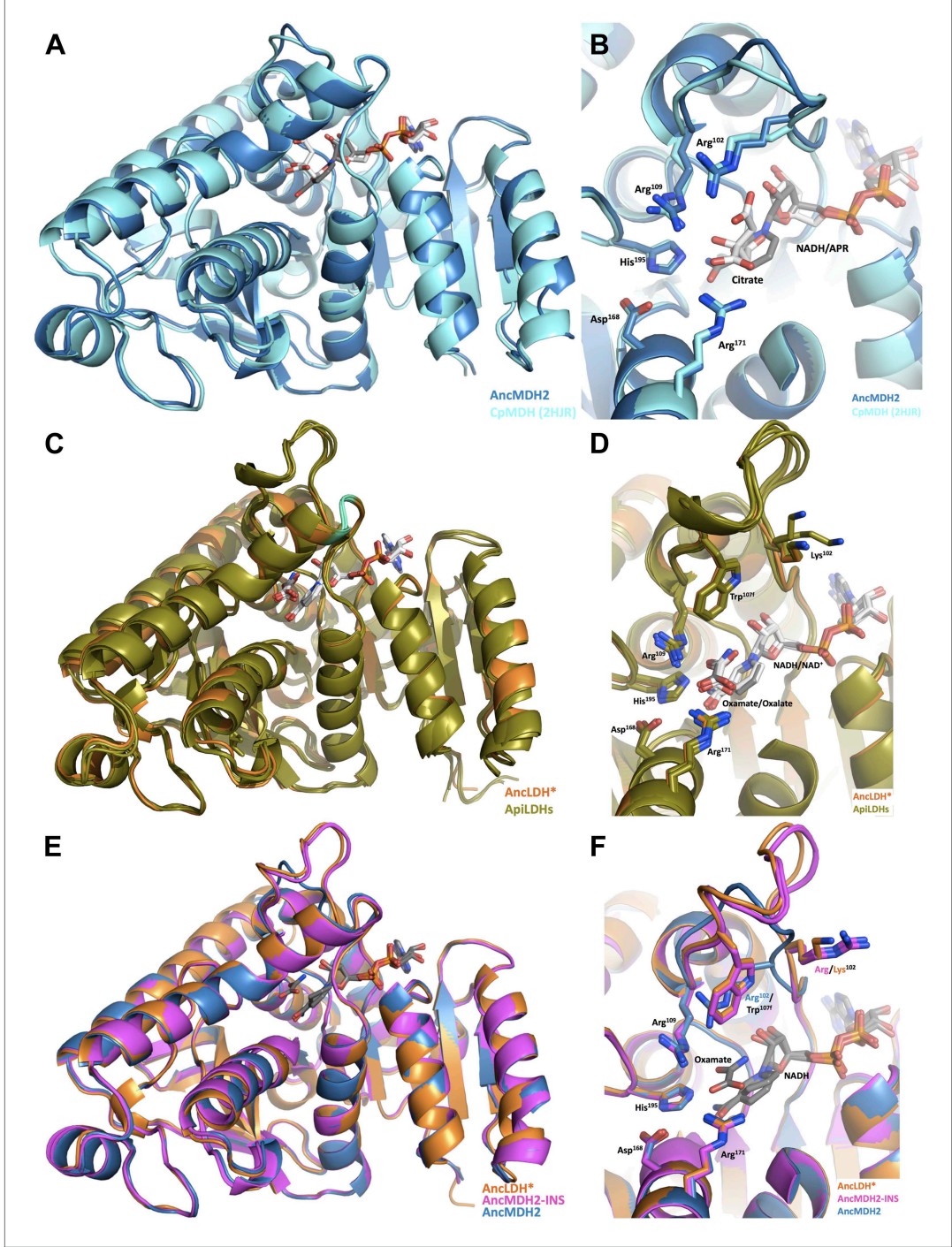

**Figure 7**. Ancestral and modern dehydrogenase structures. (**A**) Superposition of *Cp*MDH and AncMDH2. Superposition of AncMDH2 structure (blue, 4plw, chain C) and *Cp*MDH (aquamarine, 2hjr, chain A). Ligands from AncMDH2 are shown in gray; ligands from *Cp*MDH are in white. (**B**) Active site detail of *Cp* MDH and AncMDH2. Side chains of catalytic residues highlighted as sticks. (**C**) Superposition of apicomplexan LDHs and AncLDH*. Superposition of AncLDH* structure (vermilion, 4plg, chain A) and four apicomplexan LDHs (deep olive, *Pf*LDH, 1t2d, chain A, *Plasmodium berghei* (*Pb*) LDH, 1oc4, chain B, *Tg*LDH1, 1pzh, chain A, *Tg*LDH2, 1sow, chain B). Ligands from AncLDH* are shown in gray, ligands from apicomplexan LDHs are in white. The 'opposing loop' and residues 236 and 246 (discussed in text) are highlighted in cyan. (**D**) Active site detail of apicomplexan LDHs and AncLDH*. Side chains of catalytic residues highlighted as sticks. (**E**) Superposition of ancestral dehydrogenases. *Figure 7. Continued on next page*

*Figure 7. Continued*

Superposition of AncMDH2 (blue, 4plw, chain C), AncLDH* (vermilion, 4plg, chain A), and AncMDH2-INS (magenta, 4ply, chain F and 4plv, chain B). Ligands are shown in gray. (**F**) Active site detail of ancestral dehydrogenases. Side chains of catalytic residues highlighted as sticks.

The following figure supplements are available for figure 7:

**Figure supplement 1**. Crystallographic statistics table for AncLDH*, AncMDH2, AncMDH2-INS, and *Pf*LDH-W107fA structures.

The largest differences between the modern and ancestral proteins are confined to two regions: a small shift of the entire C-terminal helix, and a loop opposite the active site specificity loop (residues 242–244, hereafter called the 'opposing loop'). The modern *Plasmodium* LDHs have a two-residue deletion within this opposing loop (highlighted in cyan in *Figure 7C*), while the opposing loop is shared with AncLDH* and the *Toxoplasma* LDHs. The ancestral LDHs also share very modest oxaloacetate activity with the modern *Toxoplasma* LDHs, while the *Plasmodium* LDHs lack oxaloacetate activity (*Figure 5*). This correlation indicates the opposing loop deletion (and perhaps Ala236 and Pro246) may be responsible for the unusually strict substrate specificity of the modern *Plasmodium* LDHs.

## Ancestral malate dehydrogenase with loop insertion: AncMDH2-INS

We also crystallized AncMDH2-INS, a bifunctional AncMDH2 construct with the six-residue specificity loop insertion. This AncMDH2-INS construct represents a possible intermediate along the evolutionary trajectory between the MDH duplication event and the ancestral apicomplexan LDH. AncMDH2-INS was successfully co-crystallized with both oxamate/NADH and lactate/NADH. In the loop-closed state, the specificity loop adopts an LDH-like confirmation with Trp107f occupying the specificity position and Arg102 oriented into solution, similar to how Lys102 is positioned in the modern and ancestral LDH structures (*Figure 7F*). The lactate and oxamate structures are highly similar (RMSD ~0.2 Å), and the active site architectures are nearly indistinguishable.

The three ancestral proteins, AncMDH2, AncLDH*, and AncMDH2-INS, are all highly similar (RMSD 1.20 Å) with the main structural differences found in the conformation of the specificity loop (*Figure 7E*, RMSD ~0.9 Å excluding residues in the specificity loop). Otherwise the first shell active site residues are identical between AncMDH2-INS and AncLDH*, and the conformations of the active sites are correspondingly similar (*Figure 7F*).

## Convergent pathways available to the ancestral MDH

Given the known importance of position 102, the 'specificity residue', in substrate recognition, we wondered whether different residues at position 102 could confer pyruvate activity. Position 102 in fact differs in the four convergent LDH families: Gln in canonical LDHs (*Wilks et al., 1988*), Lys in the apicomplexan LDHs, Gly in *Cryptosporidium* LDHs (*Madern et al., 2004*), and Leu in trichomonad LDHs (*Wu and Fiser, 1999*). Could the ancestral apicomplexan MDH have evolved pyruvate specificity by any of these alternative routes? To answer this question, we evaluated the potential of these different amino acids at the 102 position to confer pyruvate specificity in the AncMDH2 background. Each mutation increases pyruvate activity, but none result in a highly specific LDH. The canonical mutation (Arg102Gln) results in the largest gain in pyruvate activity (2800-fold) and the smallest loss of oxaloacetate activity (2500-fold) (*Figure 8*). Additionally, we tested whether the full six amino acid insertion was required to confer pyruvate specificity in AncMDH2 or if simply mutating Arg102 to Trp was sufficient. The Arg102Trp mutation all but abolishes activity towards both substrates, indicating that the loop insertion was necessary to switch the specificity residue (*Figure 8*).

## Discussion

### An alternate mechanism of specificity in the convergent apicomplexan LDH

Substrate recognition in the canonical MDHs and LDHs is thought to be determined by a 'specificity residue' in the active site loop at position 102. All known MDHs have Arg at position 102, while canonical LDHs have Gln (*Chapman et al., 1999*). In the classic explanation of the molecular mechanism of

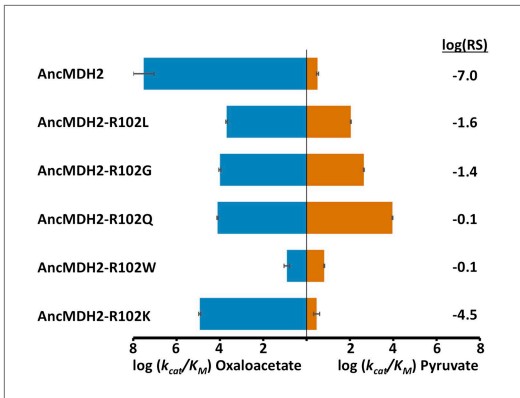

**Figure 8**. Alternative LDH mutations in AncMDH2. Relative specificity (RS) is described in legend of *Figure 4*.

The following source data are available for figure 8:

**Source data 1**. Kinetic parameters for specificity residue mutants.

substrate specificity, residue 102 discriminates between pyruvate and oxaloacetate primarily via charge conservation (*Wilks et al., 1988*). In MDHs, the positively charged Arg interacts with and balances the negatively charged carboxylate of oxaloacetate. If pyruvate were to bind in the active site, loop closure would result in a buried and unbalanced positive charge, which is unfavorable. In canonical LDHs, the neutral Gln interacts with the neutral pyruvate methyl group. Oxaloacetate binding would similarly result in the unfavorable burial of an unbalanced negative charge.

In the apicomplexan LDHs, evolution has converged on pyruvate specificity using an alternative molecular mechanism. Residue 102 is not a Gln but a positively charged Lys, similar to Arg102 of MDHs, leading many researchers to wonder why apicomplexan LDHs lack activity towards oxaloacetate (*Gomez et al., 1997*; *Dando et al., 2001*; *Winter et al., 2003*; *Brown et al., 2004*; *Kavanagh et al., 2004*; *Shoemark et al., 2007*). However, during the evolution of the apicomplexan LDH from the ancestral MDH, the six-residue insertion in the active site loop shifted both the position and identity of the 'specificity residue' from Arg102 to Trp107f. Due to the insertion, residue 102 no longer interacts with the substrate and is extruded from the active site. In contrast, the hydrophobic Trp107f packs against the C3 methyl of the pyruvate substrate. Similar to the canonical LDH, oxaloacetate binding would result in an unbalanced and buried negative charge. As a large bulky residue, Trp107f can also occlude binding of the larger oxaloacetate, in which a methylene carboxylate replaces the pyruvate methyl.

However, as discussed in detail below, this simplistic explanation is complicated by long-range epistatic interactions. When the six-residue insertion is introduced into the modern apicomplexan MDH, specificity is not switched; both specificity and activity are lost. Similarly, removal of the insertion from the modern apicomplexan LDHs fails to swap specificity and kills the enzymes. Therefore, while Trp107f is necessary for substrate specificity in the apicomplexan enzymes (as indicated by the alanine scan mutations), it is insufficient to confer specificity.

The bifunctionality of AncMDH2-INS and AncMDH2-INS-59Mut also presents a conundrum. Why do these constructs have high activity towards both pyruvate and oxaloacetate substrates? The crystal structure of AncMDH2-INS offers few clues, since the loop insertion, including Trp107f, adopts the same conformation as seen in AncLDH and the modern apicomplexan enzymes. Both the AncMDH2-INS and AncMDH2-INS-59Mut constructs have an Arg at position 102, like the MDHs. In fact, the bifunctional AncMDH2-INS-59Mut enzyme differs from the strict AncLDH by only a R102K mutation, suggesting that Arg102 is responsible for the oxaloacetate activity of AncMDH2-INS and AncMDH2-INS-59Mut. We speculate that perhaps the enzymes change conformation depending upon the substrate. When using pyruvate, these bifunctional enzymes may adopt an LDH-like conformation in which Trp107f interacts with the substrate (as seen in the crystal structure). On the other hand, when presented with oxaloacetate, perhaps Trp107f flips out of the active site, and Arg102 flips in to interact with substrate in a manner similar to the canonical MDHs. We are currently testing this hypothesis.

## Apicomplexan LDH evolved by classical neofunctionalization

Our data show that apicomplexan LDHs evolved from a horizontally transferred proteobacterial MDH by a classic neofunctionalization mechanism of gene duplication. Because debasement to a pseudogene is much more likely to occur prior to the evolution of a novel function, neofunctionalization has fallen out of favor as a mechanism for the evolution of novel functions. A variety of alternative specialization models have been proposed that feature a reduced risk of non-functionalization. Though differing in details, all specialization models feature a promiscuous common ancestor of the duplicated proteins.

The reconstructed AncMDH2, which represents the last common ancestor of the apicomplexan MDH and LDHs, is a highly active and specific MDH, preferring oxaloacetate over pyruvate by seven orders of magnitude (*Figure 6*). The activity of AncMDH2 towards pyruvate is barely detectable, requiring a high enzyme concentration to quantify. AncMDH2's $k_{cat}$ for pyruvate is 0.07 s$^{-1}$, with a $K_m$ of 20 mM, while the physiological concentration of pyruvate is estimated to be about three orders-of-magnitude lower (e.g., ~50 μM in human erythrocytes [*Garrett and Grisham, 2005*], the *Plasmodium* host during its blood stage). Based on these kinetic parameters, each AncMDH2 reduces one pyruvate molecule per hour. While the enzyme can be forced to reduce pyruvate in vitro, this negligible activity is unlikely to have been subjected to selection in vivo. Therefore, the various specialization hypotheses, which require a promiscuous ancestor, are poor models for apicomplexan LDH evolution. Activity towards pyruvate increased by over seven orders of magnitude on the evolutionary lineage between AncMDH2 and AncLDH, indicating neofunctionalization.

## A highly active, promiscuous intermediate

One of the most favored specialization models is 'escape from adaptive conflict' (EAC) (*Des Marais and Rausher, 2008*). EAC holds that functional specialization is driven by an inability to simultaneously optimize multiple functions on a single protein scaffold. Gene duplication relieves this constraint and allows for the independent optimization of conflicting functions. Although the apicomplexan AncMDH2 is highly specific, promiscuous intermediates did play a role in the functional transition between AncMDH2 and AncLDH. AncMDH2-INS and AncMDH2-INS-59Mut have high levels of MDH and LDH activity in a single protein scaffold (*Figure 6*). Both the presence of bifunctional intermediates and the high specificity of AncMDH2 conflict with fundamental predictions of the EAC specialization model.

## Convergent evolution of apicomplexan LDH involved long-range epistasis

The evolution of apicomplexan LDHs involved strong epistasis that has profoundly influenced the convergent evolution of pyruvate activity. Epistasis refers to interactions between residues that potentiate the effects of a mutation depending on the presence or absence of other residues (*Harms and Thornton, 2010*). Epistasis can constrain the order of mutations and the pathways accessible to evolution, and hence it is of great importance in understanding the evolution of novel functions. In the apicomplexan dehydrogenases, the evolutionary mutations that switched specificity from oxaloacetate to pyruvate (the six-residue insertion and Arg102Lys) are insufficient to confer pyruvate activity in modern apicomplexan MDHs (*Pf*MDH-R102K, *Pf*MDH-INS, *Cp*MDH-INS, *Pf*MDH-R102K-INS, *Figure 4*). However, these mutations are sufficient to confer pyruvate function and specificity in the AncMDH2 background (AncMDH2-INS, AncMDH2-INS-R102K, *Figure 6*). Similarly, removal of the insert from the modern LDHs (*Pf*LDH-DEL and *Tg*LDH2-DEL, *Figure 4*) kills the enzymes, while removal of the insert from the ancestral LDH (AncMDH2-R102K-59Mut, *Figure 6*) results in a weak MDH. The different effects of these mutations, depending upon the sequence of the rest of the protein, provide direct evidence of epistatic interactions.

Why do these historical mutations 'work' in the ancestral enzymes, but not in the modern ones? Epistatic interactions are often mediated by direct physical contact. For example, the active site of the ancestral MDH could have certain residues that the modern MDH lacks, residues that interact with the insertion and allow it to preferentially bind pyruvate. However, the active sites of the ancestral and modern MDHs are identical in sequence and virtually indistinguishable in structure (*Figure 7B*), as are the active sites of the ancestral and modern LDHs (*Figure 7D*) and the AncMDH2-INS intermediate (*Figure 7F*). In fact, the active sites of the MDHs and the LDHs are also identical in sequence except for the 102 position, and they are otherwise highly structurally similar. Therefore, residues remote from the active sites necessarily affect the substrate specificity of the enzymes.

In principle, these long-range epistatic residue interactions could differentially modify the structure of the active site. Certain residues found in the ancestral MDH, but not in the modern MDH, could position the active site residues so that they allowed the insertion to confer pyruvate activity. In this scenario the active site residues of the ancestral and modern MDHs would be identical, but their conformations would differ due to interactions with residues in other parts of the protein. However, the crystal structures reveal ancestral, intermediate, and modern active sites that are nearly indistinguishable, suggesting that epistasis has modified the protein dynamics or shifted the energy landscape, effects that are largely invisible to static crystal structures.

## Epistasis prevents mechanistic convergence in the LDH/MDH superfamily

Interestingly, *Bacillus subtilis* (*Bs*) LDH reverts to an MDH with only a single mutation, Gln102Arg, indicating a lack of complicating epistatic effects (*Wilks et al., 1988*). The kinetics of wild-type *Bs*LDH with pyruvate are comparable to those for the Gln102Arg mutant with oxaloacetate (e.g., *Bs*LDH has a $k_{cat}/K_M$ for pyruvate of $4.2 \times 10^6$ $M^{-1}$ $s^{-1}$, and the *Bs*LDH-Q102R mutant has the same $k_{cat}/K_M$ for oxaloacetate). However, *Bs*LDH likely is an exception in the LDH/MDH superfamily, since the reverse mutation (Arg102Gln) fails to switch specificity in MDHs from two other species (*Nicholls et al., 1992*; *Cendrin et al., 1993*). In *Haloarcula marismortui* (*Hm*) MDH, the Arg102Gln mutation switches specificity, but the mutant's $k_{cat}/K_M$ for pyruvate is 200-fold less than the wild-type's $k_{cat}/K_M$ for oxaloacetate. The Arg102Gln mutation in *Escherichia coli* (*Ec*) MDH is even less effective, as it converts a highly active MDH to an enzyme with low activity on both substrates (10,000-fold lower $k_{cat}/K_M$). Hence, the strong epistasis observed in apicomplexan LDH and MDHs is likely a general phenomenon within the superfamily.

LDH evolved convergently from MDH four separate times in the superfamily, but did the activity evolve by the same mechanism each time? Each event has resulted in a different change at the specificity residue (position 102) within the catalytic loop. However, the epistatic effects seen in the apicomplexan, *H. marismortui,* and *E. coli* dehydrogenases indicate that in general position 102 is not solely responsible for the transition from MDH to LDH. In order for the historical LDH mutations to confer pyruvate specificity, additional residues must be present to provide a permissive background (*Figure 8*). Due to the presence of different sets of permissive mutations, LDH activity has evolved from an MDH under epistatic constraints by a different mechanism four separate times.

## Large effect, gain-of-function mutation

The evolution of AncLDH from AncMDH2 involves a shift in substrate specificity by 12 orders-of-magnitude. Through the characterization of possible evolutionary intermediates, we have found that just two mutations are responsible for the great majority of this switch: the six-residue insertion and the Arg102Lys point mutation. Mutagenesis within the insertion indicates that only a single position, Trp107f, contributes strongly to pyruvate activity and specificity. Both the insertion and Arg102Lys have a large effect on preference for pyruvate vs oxaloacetate, although by differentially affecting activity towards each substrate. Incorporating the six-residue insertion into AncMDH2's substrate loop results in a 12,000-fold gain in pyruvate activity with little effect on oxaloacetate activity (*Figure 6*). Conversely, mutating Arg102 to Lys reduces oxaloacetate activity by more than 2500-fold, with minimal effect on pyruvate activity (*Figure 6*).

The apicomplexan LDH six-residue insertion is an exceptionally large gain-of-function mutation: it enhances pyruvate activity by more than four orders of magnitude while barely affecting oxaloacetate activity. In contrast, other well-studied mutations of large effect are often predominantly deleterious towards one function while modestly enhancing another. The textbook example of a gain-of-function mutation is Gln102Arg in *Bs*LDH, which causes a $10^7$-fold change in the enzyme's specificity (*Wilks et al., 1988*). The Gln102Arg mutation reduces pyruvate activity by more than 8000-fold, while enhancing activity towards oxaloacetate by only 1000-fold. Another example is given by *E. coli* isocitrate dehydrogenase (IDH), where seven mutations are necessary to switch the cofactor specificity from a 7000-fold preference for NADP to a 200-fold preference for NAD (*Chen et al., 1995*). Within this set of mutations, two reduce specificity for NADP by 6000-fold, whereas the rest enhance NAD usage 200-fold. Thus, while mutations can have both deleterious and beneficial effects on different functions, the deleterious effects typically appear greater than enhancement.

In previous ancestral sequence reconstruction studies, mutations of large effect are in fact usually loss-of-function rather than gain-of-function (e.g., RNaseA [*Jermann et al., 1995*], chymase [*Wouters et al., 2003*], and glucocorticoid receptors [*Bridgham et al., 2006*; *Ortlund et al., 2007*; *Carroll et al., 2008*, *2011*]). In these studies, the modern proteins are generally specific for one substrate, whereas the ancestral proteins are promiscuous. Furthermore, the activity of the ancestral protein is comparable to the modern descendants. Therefore, these proteins specialized by accumulating deleterious mutations, with the modern, specialized activity being the 'last function standing'. For example, the ancestral glucocorticoid receptor binds three steroid hormones tightly ($EC_{50}$ <10 nM for aldosterone, deoxycorticosterone, and cortisol), while the modern receptors bind only cortisol ($EC_{50}$ ~ 100 nM) (*Ortlund et al., 2007*). Seven historical mutations produced the modern cortisol

preference by completely eliminating aldosterone and deoxycorticosterone sensitivity yet reducing cortisol sensitivity only 50-fold. In other ancestral reconstruction studies, function-enhancing mutations have relatively minor effects, all less than a 50-fold gain in $k_{cat}/K_M$ (*Zhang and Rosenberg, 2002*; *Voordeckers et al., 2012*; *Risso et al., 2013*).

During the evolution of the malate and lactate dehydrogenase superfamily, pyruvate activity has converged multiple times despite strong constraints due to epistasis. While epistasis may constrain evolutionary options locally, there are nevertheless multiple ways to 'skin the cat' in more distant regions of protein sequence space. The apicomplexan enzymes provide a clear example of neofunctionalization in protein evolution and thereby validate the plausibility of this particular mechanism of gene duplication. Specialization mechanisms may be more common, but the evolution of novel function does not require a promiscuous genesis.

## Accession numbers

The PDB accession codes for the coordinates and structure factor files reported in this paper are 4PLC, 4PLF, 4PLG, 4PLH, 4PLT, 4PLV, 4PLW, 4PLY, and 4PLZ.

# Materials and methods

## Modern sequences

Protein sequences used in the phylogenetic analyses were identified through searches of the non-redundant database (*Pruitt et al., 2009*) with the BLASTP algorithm (*Altschul et al., 1990*) using selected query sequences. All sequences from these searches that returned BLASTP E-values <10⁻⁷ were downloaded from NCBI (www.ncbi.nlm.nih.gov). Multiple complete apicomplexan genomes (*Heiges et al., 2006*; *Gajria et al., 2008*; *Aurrecoechea et al., 2009*) were also searched for LDH and MDH homologs in order to fill out the apicomplexan portion of the tree (using a more lenient significance cutoff of E-values <10⁻⁴). Redundant sequences, synthetic constructs, and sequences from PDB files were removed. To reduce phylogenetic complexity, sequences were curated based on character length and pairwise sequence identity within each dataset (as described below).

The dataset used for the construction of the non-redundant phylogeny (*Figure 3A*) was generated using four query sequences, UniProt IDs (*UniProt Consortium, 2013*): MDHC_HUMAN, LDH_THEP1, MDHP_YEAST, and LDH6A_HUMAN. Multiple sequences were necessary to generate full coverage, due to the low sequence identity across the superfamily, which can be less than 20% between members. Sequences were removed if their character length was less than 280 or greater than 340. Limits were chosen to remove truncated/partial sequences and those featuring large insertions or terminal extensions. Sequences greater than 97% identical, determined by pairwise alignment within the dataset, were also removed. This level of identity provides a high level of detail within the tree while accelerating computational time by removing redundant taxa. The final dataset contains 1844 taxa.

Residue numbering in the text is based on the dogfish LDH convention (*Eventoff et al., 1977*) for consistency with previous work.

## Primary phylogeny construction

A multiple sequence alignment of this dataset was generated using the program MUSCLE (*Edgar, 2004*). A maximum likelihood (ML) phylogenetic tree was inferred with PhyML 3.0 (*Guindon et al., 2010*) using the LG substitution matrix (*Le and Gascuel, 2008*) and estimating the gamma parameter (12 categories) and empirical amino acid frequencies. The starting tree was generated by Neighbor-Joining (BIONJ) and searched by Nearest Neighbor Interchange (NNI); tree topology, branch lengths, and rate parameters were optimized. Branch supports were estimated with the approximate likelihood ratio test (aLRT), as implemented in PhyML, represented as either the raw aLRT statistic (roughly >8 is considered highly significant) or the confidence level that the clade is correct (*Anisimova and Gascuel, 2006*).

## Phylogeny rooting

The outgroup for rooting the L/MDH phylogeny was identified through a profile analysis of the Rossmann fold (*Rao and Rossmann, 1973*), based on a method used for OB folds and SH3 domains (*Theobald and Wuttke, 2005*). All structurally characterized Rossmann folds with 40% or less sequence identity were identified from ASTRAL SCOP 1.73 protein domain sequence database (*Chandonia et al., 2004*). Each of the 193 domains identified was searched against the SwissProt

database (*Boeckmann et al., 2003*) using BLASTP. A multiple sequence alignment for each query and SwissProt sequences with BLASTP E-values <$10^{-10}$ was created using MUSCLE. Each alignment was cropped to the limits of the original query. COMPASS (*Sadreyev et al., 2003*) was then used to generate an all-against-all scoring matrix for the 193 multiple sequence alignments. The E-values generated by COMPASS were converted to evolutionary distances as described in *Theobald and Wuttke (2005)*. A weighted least-squares phylogenetic analysis of the distance matrix was performed using PAUP (*Swofford, 2003*). First order taxon jackknifing (*Lanyon, 1985*; *Siddall, 1995*) was used to determine the robustness of tree topology, with a consensus tree calculated from all analyses.

Rossmann fold domains from α- and β-glucosidases and aspartate dehydrogenases (AspDH) were identified from the profile–profile analysis as grouping with the Rossmann fold domain from L/MDHs. An L/MDH dataset was constructed for use with the outgroup to create a rooted phylogeny. This dataset was generated by querying four sequences, UniProt IDs: MDHC_HUMAN, LDH_THEP1, MDHP_YEAST, and LDH6A_HUMAN, against the SwissProt database using BLASTP. All sequences from these searches that returned BLASTP E-values <$10^{-7}$ were downloaded from NCBI (www.ncbi.nlm.nih.gov). Redundant sequences, synthetic constructs, and sequences from PDB files were removed. Also, four taxa identified as ubiquitin-conjugating enzymes were removed due to sequence length. This SwissProt L/MDH dataset contained 595 taxa.

An outgroup dataset was constructed by querying three sequences, UniProt IDs: LICH_BACSU, AGAL_THEMA, and ASPD_THEMA, against the SwissProt database using BLASTP. All sequences from these searches that returned BLASTP E-values <$10^{-7}$ were downloaded from NCBI (www.ncbi.nlm.nih.gov). Redundant sequences, synthetic constructs, and sequences from PDB files were removed. The outgroup dataset contained 62 taxa. The SwissProt LDH, MDH, AspDH, and glucosidase datasets were combined and a multiple sequence alignment was generated using the program MUSCLE. The C-terminal domain of the glucosidases and AspDHs were removed from the MUSCLE alignment. A ML phylogenetic tree was inferred from the alignment with PhyML using the LG substitution matrix (*Whelan and Goldman, 2001*) with the gamma parameter estimated over 10 categories, no invariant sites, and estimating empirical amino acid frequencies. The initial tree was obtained by BIONJ and searched by NNI; tree topology, branch lengths, and rate parameters were optimized. Robustness of root positioning was evaluated with two truncated alignments, one with the LDH and 'LDH-like' MDH sequences removed and the other with the cytosolic and mitochondrial MDH sequences removed. Truncated alignments were input to PhyML for phylogenetic analysis using the parameters described above.

## Alternative phylogeny construction

The dataset for the alternative phylogeny (used in reconstructing alternative ancestors) is smaller and focused on apicomplexan taxa. It was generated by BLASTP searches with four query sequences, UniProt IDs: MDHC_PIG, Q76NM3_PLAF7, C6KT25_PLAF, and MDH_WOLPM for full coverage of the superfamily. Sequences were removed if their length was less than 290 or greater than 340. The dataset was culled to 60% identity, but the apicomplexan clade was filled back to 97% identity to gain resolution within the clade of interest. The final dataset contained 277 taxa. A multiple sequence alignment of this dataset was produced using the program MUSCLE. The ML tree was inferred with PhyML 3.0 using the LG substitution matrix and estimating the gamma parameter (12 categories) and empirical amino acid frequencies. The starting tree was generated by Neighbor-Joining (BIONJ) and searched by Nearest Neighbor Interchange (NNI); tree topology, branch lengths, and rate parameters were optimized.

## Ancestral sequence reconstruction

Sequences at internal nodes in phylogenies were inferred using the *codeml* program from the PAML software package (*Yang, 2007*). Posterior amino acid probabilities at each site were calculated using the LG substitution matrix, given the ML tree generated by PhyML. The initial ancestral reconstruction assumed the background amino acid frequencies implicit in the LG matrix, while the alternative reconstruction estimated background frequencies from the sequence alignment of the alternative dataset. N-/C-termini of ancestral sequences were modified manually to match those of the closest modern sequence (determined by branch length).

## Plasmid construction and mutation

*Escherichia coli* codon-optimized coding sequences were constructed for the *Plasmodium falciparum* MDH (gi#: 86171227), *Cryptosporidium parvum* MDH (gi#: 32765705), *Toxoplasma gondii* LDH1

(gi#: 237837615), *Toxoplasma gondii* LDH2 (gi#: 2497625), *Rickettsia bellii* MDH (gi#: 91205459), and ancestrally inferred protein sequences. These coding sequences were synthesized and subcloned into pET-24a, bypassing the N-terminal T7-tag but using the C-terminal 6xHis-tag. *Pf*LDH (gi#: 124513266) with six His residues added to the C-terminus was synthesized and subcloned into pET-11b. All gene synthesis and subcloning was performed by Genscript (Piscataway, NJ). All point mutations were made using the QuikChange Lightning kit from Agilent (Santa Clara, CA) and synthesized primers from IDT (Coralville, IA).

## Protein expression and purification

Plasmids were transformed in BL21 DE3 (pLysS) *E. coli* cells (Invitrogen, Grand Island, NY) for expression. Cells were grown at 37°C with 225 rpm agitation in 2xYT media supplemented with 30 mM potassium phosphate, pH 7.8 and 0.1% (wt/vol) glucose. Once cultures reached an $OD_{600}$ between 0.5–0.8, cells were induced with 0.5 mM IPTG for 4 hr. Cells were collected by centrifugation at 10,000×*g* for 15 min and stored at −80°C.

Cell pellets were thawed on ice, releasing lysozyme produced by the pLysS plasmid from within the cells, and resuspended in 15 ml lysis buffer (50 mM $NaH_2PO_4$, pH 8.0, 300 mM NaCl, 10 mM Imidazole) with 375 units of Pierce Universal Nuclease (Thermo Scientific, Rockford, IL) per 1.5 l of culture. Once homogeneously resuspended, lysate was sonicated on ice at 35% amplitude (30 s ON, 20 s OFF, 2 min total). Insoluble cell debris was separated by centrifugation at 18,000×*g* for 20 min.

Proteins were purified by nickel affinity chromatography. Clarified lysate was applied to a 5 ml HisTrap FF column (GE Healthcare, Piscataway, NJ) and eluted via an imidazole gradient from 10 mM to 500 mM on an AKTA Prime (GE Healthcare, Piscataway, NJ). Fractions were analyzed by SDS-PAGE, pooled, and concentrated using Amicon Ultracel-10 K centrifugal filters (Millipore, Billerica, MA). Finally, proteins were desalted into 50 mM Tris, pH 7.4, 100 mM NaCl, 0.1 mM EDTA and 0.01% azide by either PD10 column (GE Healthcare, Piscataway, NJ) or gel filtration over a HiPrep 16/60 Sephacryl S-200 HR column (GE Healthcare, Piscataway, NJ) on an AKTA Purifier (GE Healthcare, Piscataway, NJ). Enzyme concentrations were determined by absorbance at 280 nm, using extinction coefficients and molecular weights calculated by ExPASy's ProtParam tool (http://web.expasy.org/protparam/).

## Steady-state kinetic assays

Enzymatic reduction of pyruvate and oxaloacetate was monitored at 25°C by following the decrease in absorbance at 340 nm due to NADH oxidation on a Cary 100 Bio (Agilent, Santa Clara, CA) in 50 mM Tris, pH 7.5, 50 mM KCl. All substrates were purchased from Sigma-Aldrich (St. Louis, MO). NADH concentration was held constant at 200 µM while pyruvate/oxaloacetate concentrations were titrated. Enzyme concentrations ranged from 0.28 nM to 2.8 µM, depending on enzyme activity for a particular substrate. All experiments used 1-cm path-length quartz cuvettes with 500 µl final volume of reaction mixture.

Kinetic parameters were estimated by chi-squared fitting to either the Michaelis-Menton equation ($v/[E]_t = k_{cat} [S]/(K_M + [S])$) or a substrate inhibition equation ($v/[E]_t = k_{cat} [S]/(K_M + [S] + [S]^2/K_i)$) using the KaleidaGraph software. Three datasets were fit using a modified substrate inhibition equation with $K_M = K_i$ for identifiability and to prevent the $K_i$ being less than $K_M$. These datasets were: AncMDH2-INS-59Mut oxaloacetate and AncMDH2-R102Q for both oxaloacetate and pyruvate. Kinetic constants $k_{cat}$, $K_M$, and $k_{cat}/K_M$ are consistently reported in units of $s^{-1}$, M, and $s^{-1}M^{-1}$, respectively.

Aqueous oxaloacetate spontaneously decarboxylates to pyruvate at 25°C and neutral pH at a rate of ~$3 \times 10^{-5}$ $s^{-1}$ (approximately 10% per hr) (*Wolfenden et al., 2011*). As a result, oxaloacetate preparations contain appreciable pyruvate contamination (approximately 1–3% from Sigma-Aldrich, depending on batch) and must be handled with care. All oxaloacetate stock solutions were made fresh before each assay and kept on ice to keep decarboxylation to a minimum. For enzymes with low pyruvate activity, the oxaloacetate decarboxylation has a negligible affect on measured rates. However, enzymes with appreciable pyruvate activity can display an apparent, artifactual oxaloacetate activity that is due to pyruvate contamination (*Parker and Holbrook, 1981*; *Wilks et al., 1988*; *Shoemark et al., 2007*). In this work, seven such proteins are *Pf*LDH, *Pf*LDH-K102R, *Tg*LDH1, *Tg*LDH2, AncLDH, AncLDH*, and AncMDH2-INS-R102K. For these proteins, oxaloacetate activity was assayed at high enzyme

concentration (600 nM–1 µM), resulting in a biphasic $\Delta A_{340}$ trace with an initial burst in which pyruvate is rapidly consumed followed by a slower linear phase representing oxaloacetate reduction. The post-burst (slow) phase of the $\Delta A_{340}$ trace was used to quantify the oxaloacetate catalytic rate (*Parker and Holbrook, 1981*; *Wilks et al., 1988*). This procedure controls for the standing pyruvate contamination but does not account for the relatively slow spontaneous decarboxylation during the assay. Hence, the oxaloacetate $k_{cat}/K_m$ values for the seven enzymes with high pyruvate activity should be considered upper limits on the true oxaloacetate activity. The low or negligible oxaloacetate activities of these seven enzymes were further confirmed by undetectable malate/$NAD^+$ reactions in spectroscopic steady state enzyme assays, and the absence of malate product as determined from 1D proton NMR (3 µM enzyme, 5 mM oxaloacetate, 5 mM NADH in $NaCl/P_i/D_2O$ pH 7.5 over four hour reaction) (*Shoemark et al., 2007*).

## Protein crystallization

Crystallization trials were conducted by hanging-drop vapor-diffusion at room temperature using Crystal Screen and Crystal Screen 2 from Hampton Research (Aliso Viejo, CA). Drops consisting of 2 µl reservoir solution and 2 µl protein stock were equilibrated against 1 ml of reservoir solution. Crystals of the ancestral proteins were identified from condition #43 of Crystal Screen (30% (wt/vol) polyethylene glycol 1500) and further refined by adding 0.1 M sodium HEPES.

Crystals of the ternary complexes were grown at room temperature by hanging-drop vapor-diffusion with 4 µl drops of 1:1 precipitating buffer:protein. AncMDH2 (25 mg/ml) was co-crystallized with 2 mM oxamate/NADH in 35% (wt/vol) PEG-1500, 0.1 M sodium HEPES, pH 7.5 and with 2 mM L-lactate/NADH in 30% (wt/vol) PEG-1500, 0.1 M sodium HEPES, pH 7.3. AncMDH2-INS (18 mg/ml) was co-crystallized with 1 mM oxamate/NADH and 1 mM L-lactate/NADH in 25% (wt/vol) PEG-1500, 0.1 M sodium HEPES, pH 8.1. AncLDH* (20 mg/ml) was co-crystallized with 2 mM oxamate/NADH and 2 mM L-lactate/NADH in 20% (wt/vol) PEG-1500, 0.1 M sodium HEPES, pH 7.5. *Pf* LDH_W107fA (20 mg/ml) was co-crystallized with 1.2 mM oxamate/2 mM NADH in 22% (wt/vol) PEG-1000.

All crystals were cryoprotected with a 30% (wt/vol) dextrose solution (15 mg dextrose dissolved in 50 µl reservoir solution). Crystals were harvested from the drop, soaked in 15% (wt/vol) dextrose solution for 3 min, transferred to the 30% solution, and flash-frozen immediately in liquid $N_2$.

## Structure determination

Diffraction datasets were collected at the SIBYLS beamline (12.3.1, Lawrence Berkeley National Laboratory, Berkeley, CA). All datasets were indexed, integrated, and scaled with XDS/XSCALE (*Xds, 2010*). Datasets included all reflections that were significant according to the CC(1/2) criterion (flagged as '*' in the XDS output), which typically extended to much weaker data than the conventional 2 sigma cutoff (*Karplus and Diederichs, 2012*; *Diederichs and Karplus, 2013*). The resolution of the data is defined as the resolution bin where CC(1/2) = 0.5. Structures were solved by molecular replacement using AutoMR in PHENIX (*Adams et al., 2010*). Homology models for the AncMDH2 and AncLDH* datasets were generated by the Phyre2 server (*Kelley and Sternberg, 2009*). The AncMDH2 homology model was based on the structure for *Cryptosporidium parvum* MDH (2hjr, 62% sequence identity, *Vedadi et al., 2007*), while the model for AncLDH* was based on the *Toxoplasma gondii* LDH1 structure (1pzf, 65% sequence identity, *Kavanagh et al., 2004*). AncMDH2-INS datasets were solved using the AncMDH2 structure as the model. *Pf* LDH_W107fA dataset was solved using the *P. falciparum* LDH structure (1t2d) structure as a model. All models were improved by rounds of manual building in Coot (*Emsley et al., 2010*) and refinement by phenix.refine in PHENIX. Model quality of all structures was validated with MolProbity (*Davis et al., 2007*; *Chen et al., 2010*) in PHENIX. All structural alignments were generated using THESEUS (*Theobald and Wuttke, 2008*). Structure images were rendered with PyMOL.

In all structure models, the enzyme is generally found in either of two states, the loop-open state or the loop-closed state, and often both states are observed in different monomers (chains) in the asymmetric unit. Loop-open states are generally more disordered (with higher B-factors) and sometimes the chain could not be traced for the loop. In loop-open states the substrate typically also has weak electron density and cannot be fit reliably. In loop-closed states the substrate generally has strong electron density, but complications arose for proteins co-crystallized with lactate and malate.

NADH will eventually oxidize to NAD+ spontaneously in the crystals and during data collection due to oxidative radiation damage. Malate can also oxidize to oxaloacetate in the crystals, and oxaloacetate is expected to spontaneously decarboxylate to pyruvate relatively quickly (over the multi-week time frame of crystallization). Additionally, pyruvate spontaneously reacts with NAD+ to form a covalent pyruvate-NAD adduct (*White et al., 1976*). Although we added NADH to the crystallization mixture, we cannot reliably determine whether NADH, NAD+, or a mixture of the two is found in the crystals, and we have variable evidence for the pyruvate-NAD adduct. In both of the structures crystallized with malate (4plc and 4ply), the substrate electron density and loop conformation indicated that the malate had converted to pyruvate and/or lactate. Therefore, for the enzymes crystallized with lactate and malate, the fine details of substrate conformation and identity in the models should be viewed with skepticism, as we are fairly confident we have an unresolvable mixture of pyruvate, lactate, and pyruvate-NAD adduct in the active site, all of which will produce similar electron density. These concerns with substrate do not apply to the oxamate models (although the uncertainty in cofactor state still remains). Finally, in the 4plw model (AncMDH2 crystallized with lactate), clear phosphates were seen in the active site instead of lactate, presumably carried through from the initial purification step.

## Acknowledgements

This work was supported by the National Institutes of Health, NIH grants R01GM096053 and R01GM094468. The crystallographic data collection was conducted at the SIBYLS beamline at the Advanced Light Source (ALS), a national user facility operated by Lawrence Berkeley National Laboratory on behalf of the Department of Energy, Office of Basic Energy Sciences, through the Integrated Diffraction Analysis Technologies (IDAT) program, supported by DOE Office of Biological and Environmental Research. Additional support comes from the National Institute of Health project MINOS (R01GM105404). We would also like to thank Chris Miller, Phillip Steindel, and Catherine Theobald for critical commentary on the manuscript.

## Additional information

### Funding

| Funder | Grant reference number | Author |
| --- | --- | --- |
| National Institutes of Health | R01GM096053 | Douglas L Theobald |
| National Institutes of Health | R01GM094468 | Douglas L Theobald |
| National Institutes of Health | R01GM105404 | Scott Classen |

The funders had no role in study design, data collection and interpretation, or the decision to submit the work for publication.

### Author contributions

JIB, DLT, Conception and design, Acquisition of data, Analysis and interpretation of data, Drafting or revising the article; JRJ, BCB, Acquisition of data, Analysis and interpretation of data; SC, Acquisition of data, Analysis and interpretation of data, Drafting or revising the article

## Additional files

### Supplementary files

• Supplementary file 1. **Sequences, alignments, and trees**. Alignments and tree files for both the original (*Figure 3*) and the alternative phylogeny. Alignment for *Figure 5—figure supplement 1*. Ancestral FASTA files and posterior probabilities for each ancestral sequence (parsed in *Figure 6—figure supplements 1–6*).

• Supplementary file 2. **Molecular weights and extinction coefficients**. ExPASy calculated molecular weights and extinction coefficients for all proteins used within this study.

## Major dataset

The following datasets were generated:

| Author(s) | Year | Dataset title | Dataset ID and/or URL | Database, license, and accessibility information |
|---|---|---|---|---|
| Boucher JI, Jacobowitz JR, Beckett BC, Classen S, Theobald DL | 2014 | Crystal structure of ancestral apicomplexan lactate dehydrogenase with malate | http://www.pdb.org/pdb/explore/explore.do?structureId=4plc | Publicly available at RCSB Protein Data Bank. |
| Boucher JI, Jacobowitz JR, Beckett BC, Classen S, Theobald DL | 2014 | Crystal structure of ancestral apicomplexan lactate dehydrogenase with lactate | http://www.pdb.org/pdb/explore/explore.do?structureId=4plf | Publicly available at RCSB Protein Data Bank. |
| Boucher JI, Jacobowitz JR, Beckett BC, Classen S, Theobald DL | 2014 | Crystal structure of ancestral apicomplexan lactate dehydrogenase with oxamate | http://www.pdb.org/pdb/explore/explore.do?structureId=4plg | Publicly available at RCSB Protein Data Bank. |
| Boucher JI, Jacobowitz JR, Beckett BC, Classen S, Theobald DL | 2014 | Crystal structure of ancestral apicomplexan malate dehydrogenase with oxalate | http://www.pdb.org/pdb/explore/explore.do?structureId=4plh | Publicly available at RCSB Protein Data Bank. |
| Boucher JI, Jacobowitz JR, Beckett BC, Classen S, Theobald DL | 2014 | Crystal structure of ancestral apicomplexan malate dehydrogenase with oxamate | http://www.pdb.org/pdb/explore/explore.do?structureId=4plt | Publicly available at RCSB Protein Data Bank. |
| Boucher JI, Jacobowitz JR, Beckett BC, Classen S, Theobald DL | 2014 | Crystal structure of ancestral apicomplexan malate dehydrogenase with lactate | http://www.pdb.org/pdb/explore/explore.do?structureId=4plv | Publicly available at RCSB Protein Data Bank. |
| Boucher JI, Jacobowitz JR, Beckett BC, Classen S, Theobald DL | 2014 | Crystal structure of ancestral apicomplexan malate dehydrogenase with lactate | http://www.pdb.org/pdb/explore/explore.do?structureId=4plw | Publicly available at RCSB Protein Data Bank. |
| Boucher JI, Jacobowitz JR, Beckett BC, Classen S, Theobald DL | 2014 | Crystal structure of ancestral apicomplexan malate dehydrogenase with malate | http://www.pdb.org/pdb/explore/explore.do?structureId=4ply | Publicly available at RCSB Protein Data Bank. |
| Boucher JI, Jacobowitz JR, Beckett BC, Classen S, Theobald DL | 2014 | Crystal structure of Plasmodium falciparum lactate dehydrogenase mutant W107fA | http://www.pdb.org/pdb/explore/explore.do?structureId=4plz | Publicly available at RCSB Protein Data Bank. |

The following previously published datasets were used:

| Author(s) | Year | Dataset title | Dataset ID and/or URL | Database, license, and accessibility information |
|---|---|---|---|---|
| Vedadi M, Lew J, Artz J, Amani M, Zhao Y, Dong A, Wasney GA, Gao M, Hills T, Brokx S, Qiu W, Sharma S, Diassiti A, Alam Z, Melone M, Mulichak A, Wernimont A, Bray J, Loppnau P, Plotnikova O, Newberry K, Sundararajan E, Houston S, Walker J, Tempel W, Bochkarev A, Kozieradzki I, Edwards A, Arrowsmith C, Roos D, Kain K, Hui R | 2007 | Crystal Structure of Cryptosporidium parvum malate dehydrogenase | http://www.pdb.org/pdb/explore/explore.do?structureId=2hjr | Publicly available at RCSB Protein Data Bank. |
| Kavanagh KL, Elling RA, Wilson DK | 2004 | T.gondii LDH1 ternary complex with APAD+ and oxalate | http://www.pdb.org/pdb/explore/explore.do?structureId=1pzf | Publicly available at RCSB Protein Data Bank. |
| Cameron A, Read J, Tranter R, Winter VJ, Sessions RB, Brady RL, Vivas L, Easton A, Kendrick H, Croft SL, Barros D, Lavandera JL, Martin JJ, Risco F, Garcia-Ochoa S, Gamo FJ, Sanz L, Leon L, Ruiz JR, Gabarro R, Mallo A, De Las Heras FG | 2004 | Plasmodium falciparum lactate dehydrogenase complexed with NAD+ and oxalate | http://www.pdb.org/pdb/explore/explore.do?structureId=1t2d | Publicly available at RCSB Protein Data Bank. |

| Winter VJ, Cameron A, Tranter R, Sessions RB, Brady RL | 2003 | Lactate dehydrogenase from Plasmodium berghei | http://www.pdb.org/pdb/explore/explore.do?structureId=1oc4 | Publicly available at RCSB Protein Data Bank. |
|---|---|---|---|---|
| Rothery EL, Mowat CG, Miles CS, Walkinshaw MD, Reid GA, Chapman SK | 2003 | H61M mutant of flavocytochrome c3 | http://www.pdb.org/pdb/explore/explore.do?structureId=1p2h | Publicly available at RCSB Protein Data Bank. |
| Kavanagh KL, Wilson DK | | T. gondii bradyzoite-specific LDH (LDH2) in complex with NAD and oxalate | http://www.pdb.org/pdb/explore/explore.do?structureId=1sow | Publicly available at RCSB Protein Data Bank. |

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
