## [Decision Letter]

Thank you for sending your work entitled “An atomic-resolution view of neofunctionalization in the evolution of apicomplexan lactate dehydrogenases” for consideration at *eLife.* Your article has been favorably evaluated by a Senior editor, a Reviewing editor, and 2 reviewers.

The Reviewing editor and the reviewers discussed their comments before we reached this decision. If you have any questions about prioritizing changes in response to these comments, please do not hesitate to contact us for advice.

Please examine these comments with great care. They come from two preeminent experts and both reviewers will need to be satisfied with your revised manuscript.

Reviewer #1:

Boucher et al explore the genesis and evolution of apicomplexan lactate dehydrogenase (LDH) by gene duplication of a malate dehydrogenase (MDH) followed by a shift in substrate specificity. They combine phylogeny, sequence analysis and ancestral reconstruction with protein crystallography and enzymatic assays to determine sequences underlying substrate specificity and possible evolutionary paths for the shift in function. This multi-functional approach provides new insights into the shift of enzyme function in this case, and may be relevant to the evolution of other enzymes.

The omission of the underlying data for the sequence analysis – protein sequences, alignments, trees, and ancestral sequences – severely detract from the paper, both as an overall advance in understanding of this enzyme evolution and for the general method, as well as for my ability to review it. I strongly urge that these data be provided in a useful manner (i.e. machine-readable text files, not pdfs).

The paper is well written from an English language and grammatical point of view, but I found the flow of the arguments to be difficult to follow, frequently overstated, and not always cohesive.

MDH/LDH Phylogeny Analysis:

The breadth of the phylogenetic analysis is not clear. The use of several starting proteins for Blast analysis, and a tight cutoff rather than psi-blast or HMMs, cast some doubt on how comprehensive the survey of MDH/LDH homologs is.

The phylogenetic analysis would greatly benefit from an explicit discussion of the species distributions. Are the mitochondrial and cytosolic MDHs unique to and universal across free-living eukaryotes? Where do bacterial and archaeal MDHs fall? Are the LDHs also found in all free-living eukaryotes? Why was LDH re-invented in apicomplexans and Trichomonas - do they also have a canonical LDH, or as obligate parasites, did their ancestors lose it? Can glycolysis still proceed and lactate be excreted in the absence of LDH? The answers to these questions are important in providing an evolutionary and functional context for the evolution of these new LDH genes and the possible functionality of ancestral enzymes of relaxed specificity.

The text states that “α/β- glucosidases and aspartate dehydrogenases as outgroups”, but only a single α-glucosidase is shown in the figure. According to SCOP, the MDH/LDH and α/β- glucosidases share a superfamily, but the aspDH are in a separate superfamily. These discrepancies should be resolved.

The only support for the hypothesis that LDHs derived from “LDH-like MDHs” is the statement in the figure legend that this branch has an aLRT support of 42. Please explain this measurement and why it supports the claim of 4 inventions of LDH from MDH, and document the use of the aLRT in the Methods. Certainly these LDHs cannot be said to be “deep within MDH clades”, unlike the other cases. Likewise, the claim that the rooting of the tree divides it into two major clades is not quantitatively supported.

Sequence Analysis and Ancestral Sequence Reconstruction:

The authors oversimplify the initial comparison of apicomplexan LDH and MDH proteins, focusing on the 5-AA insert (confusingly referred to as a “6-residue loop insertion”), resulting in an awkward backtrack in later sections, where they are forced to consider additional changes between sequences. A more complete analysis of the substantial changes between Apicomplexan MDH and LDH proteins at the beginning would be far more helpful, informative, and clear. This should include mention of position 103, which is within the loop region and is substantially different in LDH (small hydrophobic) from MDH (basic). It is not surprising that an alanine scan would have little effect on this position; a mutation to the 'ancestral' lysine would be more informative. Since it is adjacent to K102 and of the same charge, it is more likely than not that this change influences either substrate specificity or enzyme activity.

The claim that Trp107f is the “modern apicomplexan specificity residue” is a bit overblown. It is the most important single residue to determine specificity and activity within the context of the insertion and other aspects of apicomplexan LDH, but it is unhelpful to strongly analogize with the traditional specificity residue at #102. Its molecular mechanism is not described and the uniqueness of its importance is also not resolved. Supplemental Figure 4 lacks any legend and I find it difficult to interpret.

The epistatic interactions seem to be over-interpreted. For instance, there is a discussion section headed “Convergent evolution of apicomplexan LDH involved long-range epistasis” but this is only inferred from the lack of obvious short-range epistasis, and no mechanistic model is presented for what this epistasis might be. Similarly, the conclusion that epistatic interactions are common simply because residue 102 mutations do not always result in strong substrate switching is not warranted; for epistasis to be shown, an interaction between two mutations must be directly shown.

The claims that: “there are no clear examples of classic neofunctionalization or gain-of-function mutations” and “the apicomplexan enzymes provide the first clear example of neofunctionalization in protein evolution” is unsupported. Cited previous examples include RNaseA, an antifreeze protein in Antarctic zoarcid fish (PMID: 21115821), and retinoic acid receptors (PMID: 16839186). It could also be argued that since the MDH/LDH system simply involves changing substrate specificity, that the change is not particularly 'neo' but rather a tuning of an existing function.

Reviewer #2:

In this interesting manuscript, the authors describe a well-supported case of neofunctionalization after gene duplication. The authors chose a unique enzyme that is a malaria drug target and convincingly demonstrate that it gained a new function after originating by gene duplication of an enzyme horizontally transferred from bacteria. Interestingly, this new function is not only essential for these protist parasites, but also has an analogous (independently evolved, convergent) counterpart in other organisms.

I think this might become a textbook example of neofunctionalization, because it is so thoroughly supported by ample evidence, from computation to biochemistry and crystallography. As such, I think it would make the paper more attractive, if some of the material is summarized better. E.g., a summary “graphical abstract” figure that shows the logic of the project, and tells the whole story at a glance might be great to have. Currently, the key important points drown in the minutiae of rationalizations and experimental details. Not that the manuscript is poorly written. I think it is very good, and the subsection title lines are particularly well-crafted. However, some further thinking might be in order to increase the impact of the paper and make it more accessible to a wider audience. Also, it would be great to articulate and summarize explicitly the mechanisms that confer specificity in 3 enzymes: MDH and two convergent LDHs. E.g., “characterized by long-range epistasis, a promiscuous intermediate, and relatively few gain-of-function mutations of large effect” is an excellent summary, but it does not help much in terms of mechanistic understanding. I would love to see (not necessarily in the Abstract) equally crisp mechanistic summary. The beauty of this work is in gaining so many insights into the mechanism, and I think it should be highlighted better.

On a technical side, I am impressed with the crystallography and the role of structures in the project. It should be noted, however, that ancestral reconstruction from highly divergent present-day sequences is a very difficult problem and is prone to many types of errors. The authors should be commended on exploring alternative reconstructions. Also, along the lines that final results justify the means, the authors show that “resurrected” proteins are very specific to their expected substrates. However, I would like to see more discussion and thought devoted to uncertainty with reconstruction, and zooming in the regions and substitutions that confer specificity. I am not suggesting any additional experiments – what the authors did looks sufficient – but simply giving a reader better appreciation that ancestral reconstruction is always highly hypothetical and is never truly reliable might be useful. Is it at all conceivable that the ancestral enzyme had broader specificity, but reconstruction methods were overwhelmed by the number of bacterial MDHs or what not? Nature is there to fool you.

---

## [Author Response]

Reviewer #1:

*Boucher et al explore the genesis and evolution of apicomplexan lactate dehydrogenase (LDH) by gene duplication of a malate dehydrogenase (MDH) followed by a shift in substrate specificity. They combine phylogeny, sequence analysis and ancestral reconstruction with protein crystallography and enzymatic assays to determine sequences underlying substrate specificity and possible evolutionary paths for the shift in function. This multi-functional approach provides new insights into the shift of enzyme function in this case, and may be relevant to the evolution of other enzymes*.

*The omission of the underlying data for the sequence analysis -protein sequences, alignments, trees, and ancestral sequences – severely detract from the paper, both as an overall advance in understanding of this enzyme evolution and for the general method, as well as for my ability to review it. I strongly urge that these data be provided in a useful manner (i.e. machine-readable text files, not pdfs)*.

We agree, apologize for the oversight, and now provide this important additional data and information, including all sequences, alignments, and trees in plain text.

MDH/LDH Phylogeny Analysis:

*The breadth of the phylogenetic analysis is not clear. The use of several starting proteins for Blast analysis, and a tight cutoff rather than psi-blast or HMMs, cast some doubt on how comprehensive the survey of MDH/LDH homologs is*.

We want to emphasize that the purpose of our paper is not to provide a comprehensive phylogenetic analysis of all MDH/LDH homologs, as this has been done previously by others ([55], 2004; [38]; Zhou and Keithly 2002). Our analysis, though much broader than these studies, largely supports the major conclusions from the earlier work. Rather, we focus on the Apicomplexan homologs, esp. the convergent LDHs, which have received less phylogenetic attention.

*The phylogenetic analysis would greatly benefit from an explicit discussion of the species distributions. Are the mitochondrial and cytosolic MDHs unique to and universal across free-living eukaryotes? Where do bacterial and archaeal MDHs fall? Are the LDHs also found in all free-living eukaryotes? Why was LDH re-invented in apicomplexans and Trichomonas – do they also have a canonical LDH, or as obligate parasites, did their ancestors lose it? Can glycolysis still proceed and lactate be excreted in the absence of LDH? The answers to these questions are important in providing an evolutionary and functional context for the evolution of these new LDH genes and the possible functionality of ancestral enzymes of relaxed specificity*.

We have now added a few sentences addressing each of these questions in the main text (“Results: LDH enzymes have evolved independently at least four times”), including a new supplemental figure highlighting the bacterial, eukaryotic, and archaeal distribution on the global tree (Figure 3—figure supplement 2).

*The text states that “α/β- glucosidases and aspartate dehydrogenases as outgroups”, but only a single α-glucosidase is shown in the figure. According to SCOP, the MDH/LDH and α/β- glucosidases share a superfamily, but the aspDH are in a separate superfamily. These discrepancies should be resolved*.

As explained in the Methods, the closest Rossmann folds were used to root the tree, as determined from a profile-based phylogenetic analysis. Apparently, only the Rossmann fold domains of the MDH/LDHs and the aspDHs are homologous.

*The only support for the hypothesis that LDHs derived from “LDH-like MDHs” is the statement in the figure legend that this branch has an aLRT support of 42. Please explain this measurement and why it supports the claim of 4 inventions of LDH from MDH, and document the use of the aLRT in the Methods. Certainly these LDHs cannot be said to be “deep within MDH clades”, unlike the other cases. Likewise, the claim that the rooting of the tree divides it into two major clades is not quantitatively supported*.

We have expanded the section on the rooting, and we now also explain the aLRT branch support statistic (developed by Olivier Gascuel and colleagues, and implemented in PhyML), and show the supports on the tree in Figure 3—figure supplement 1. Briefly, the aLRT is an approximate log-likelihood ratio statistic, where raw values > 8 are roughly “statistically significant”, representing an e^8=3000-fold better fit of the tree to the data when the clade is present (in terms of the likelihood). The aLRT can also be used in a traditional significance test to provide a confidence level (between 0 and 1, similar to a probability) for a clade. While the exact ML position of the root is poorly supported, there is very high support for a root position within the MDH portion of the tree, somewhere near the center.

Sequence Analysis and Ancestral Sequence Reconstruction:

*The authors oversimplify the initial comparison of apicomplexan LDH and MDH proteins, focusing on the 5-AA insert (confusingly referred to as a “6-residue loop insertion”), resulting in an awkward backtrack in later sections, where they are forced to consider additional changes between sequences. A more complete analysis of the substantial changes between Apicomplexan MDH and LDH proteins at the beginning would be far more helpful, informative, and clear*.

We have expanded the description of the differences between the modern MDH and LDH proteins.

*This should include mention of position 103, which is within the loop region and is substantially different in LDH (small hydrophobic) from MDH (basic). It is not surprising that an alanine scan would have little effect on this position; a mutation to the 'ancestral' lysine would be more informative. Since it is adjacent to K102 and of the same charge, it is more likely than not that this change influences either substrate specificity or enzyme activity*.

We originally thought that the 103 position was unlikely to be responsible for differences in activity/specificity between the apicomplexan MDH and LDHs. Our reasoning was based on the following structural evidence. In the crystal structures, the MDH Lys103 points in the opposite direction of Arg102, away from the active site, not contacting the substrate; (the positively charged nitrogens in the two sidechains are over 11 angstroms distant). Similarly, the beta carbon of the LDH 103 residue (Ala,Val,Leu, or Ile) points away from the active site into solution, in the opposite direction of Lys102, at the tip of the insertion in the loop, relatively far from active site and substrate. Nevertheless, data always trumps words, so at the reviewer’s suggestion we made the A103K mutation in both the modern *Pf*LDH and in the ancestor. Both mutants are kinetically indistinguishable from WT for both pyruvate and oxaloacetate, confirming our original inclination.

*The claim that Trp107f is the “modern apicomplexan specificity residue” is a bit overblown. It is the most important single residue to determine specificity and activity within the context of the insertion and other aspects of apicomplexan LDH, but it is unhelpful to strongly analogize with the traditional specificity residue at #102. Its molecular mechanism is not described and the uniqueness of its importance is also not resolved*.

As the reviewer agrees, our data indicates that Trp107f residue is “the most important single residue to determine specificity and activity within the context of the insertion and other aspects of the apicomplexan LDH” – hence our assertion that Trp107f is the apicomplexan “specificity residue” (a convenient yet simplistic label). Clearly Trp107f is not the only residue important for specificity – a major point of our paper – but the same is true of the “traditional” specificity residue at position 102, which is at best necessary but insufficient for determining substrate specificity.

The molecular mechanism by which apicomplexan Trp107f contributes to pyruvate recognition likely involves the hydrophobic Trp sidechain interacting with the hydrophobic pyruvate methyl and the bulky Trp sidechain sterically occluding the methylene carboxylate of oxaloacetate. We have now added a sentence to this effect in the text.

*Supplemental*
Figure 4
*lacks any legend and I find it difficult to interpret*.

We now provide a legend for Figure 4. The main point is that the W107fA mutant is highly similar to the WT, except for the active site loop which is partially disordered and in the open conformation.

*The epistatic interactions seem to be over-interpreted. For instance, there is a discussion section headed “Convergent evolution of apicomplexan LDH involved long-range epistasis”but this is only inferred from the lack of obvious short-range epistasis, and no mechanistic model is presented for what this epistasis might be*.

Our conclusion of long-range epistasis is not based on negative evidence. Rather, epistasis is defined as a mutation having different effects depending on the identity of residues at other sites. Our data provide clear evidence of this; both the R102K mutation and the loop insertion have differing kinetic effects when introduced in the modern vs ancestral proteins. The residues that comprise the active sites of the modern and ancestral proteins are identical; the residue differences are found only at sites removed from the active site. Hence, the epistasis we observe is necessarily long-range. Elucidating the precise mechanism of this long-range epistasis will not be trivial, but it is currently one of our major ongoing projects. We suspect that the mechanism will come down to shifting energy landscapes (a type of protein dynamics).

*Similarly, the conclusion that epistatic interactions are common simply because residue 102 mutations do not always result in strong substrate switching is not warranted; for epistasis to be shown, an interaction between two mutations must be directly shown*.

Residue 102 mutations frequently result in different levels of substrate specificity, depending on which protein they are made in. This is direct, *prima facie* evidence of epistasis, by definition. Epistatic interactions are not restricted to two sites, but can exist between different sets of sites.

*The claims that: “there are no clear examples of classic neofunctionalization or gain-of-function mutations” and “the apicomplexan enzymes provide the first clear example of neofunctionalization in protein evolution” is unsupported. Cited previous examples include RNaseA, an antifreeze protein in Antarctic zoarcid fish (PMID: 21115821), and retinoic acid receptors (PMID: 16839186). It could also be argued that since the MDH/LDH system simply involves changing substrate specificity, that the change is not particularly 'neo' but rather a tuning of an existing function*.

We agree with the reviewer that there are other potential examples of neo-functionalization, and that what counts as a new function is largely a matter of degree. We have therefore changed “the first clear example” to “a clear example”.

Reviewer #2:

*In this interesting manuscript, the authors describe a well-supported case of neofunctionalization after gene duplication. The authors chose a unique enzyme that is a malaria drug target and convincingly demonstrate that it gained a new function after originating by gene duplication of an enzyme horizontally transferred from bacteria. Interestingly, this new function is not only essential for these protist parasites, but also has an analogous (independently evolved, convergent) counterpart in other organisms*.

*I think this might become a textbook example of neofunctionalization, because it is so thoroughly supported by ample evidence, from computation to biochemistry and crystallography. As such, I think it would make the paper more attractive, if some of the material is summarized better. E.g. a summary “graphical abstract” figure that shows the logic of the project, and tells the whole story at a glance might be great to have. Currently, the key important points drown in the minutiae of rationalizations and experimental details. Not that the manuscript is poorly written. I think it is very good, and the subsection title lines are particularly well-crafted. However, some further thinking might be in order to increase the impact of the paper and make it more accessible to a wider audience*.

We have made a new graphic that succinctly summarizes the mutation experiments. We hope that Figure 5 already provides a good graphical summary of the phylogenetic and activity data.

*Also, it would be great to articulate and summarize explicitly the mechanisms that confer specificity in 3 enzymes: MDH and two convergent LDHs. E.g., “characterized by long-range epistasis, a promiscuous intermediate, and relatively few gain-of-function mutations of large effect” is an excellent summary, but it does not help much in terms of mechanistic understanding. I would love to see (not necessarily in the Abstract) equally crisp mechanistic summary. The beauty of this work is in gaining so many insights into the mechanism, and I think it should be highlighted better*.

We have now added a couple sentences regarding the mechanism to the Abstract, and we have also added a new section in the Discussion that goes into this in detail (“An alternate mechanism of specificity in the convergent apicomplexan LDH”).

*On a technical side, I am impressed with the crystallography and the role of structures in the project. It should be noted, however, that ancestral reconstruction from highly divergent present-day sequences is a very difficult problem and is prone to many types of errors. The authors should be commended on exploring alternative reconstructions. Also, along the lines that final results justify the means, the authors show that “resurrected” proteins are very specific to their expected substrates. However, I would like to see more discussion and thought devoted to uncertainty with reconstruction, and zooming in the regions and substitutions that confer specificity. I am not suggesting any additional experiments – what the authors did looks sufficient – but simply giving a reader better appreciation that ancestral reconstruction is always highly hypothetical and is never truly reliable might be useful. Is it at all conceivable that the ancestral enzyme had broader specificity, but reconstruction methods were overwhelmed by the number of bacterial MDHs or what not? Nature is there to fool you*.

We have expanded the section on the alternate reconstructions.